

# Comparing and validating intra-farm and farm-to-farm wakes across different mesoscale and high-resolution wake models

Jana Fischereit[1], Kurt Schaldemose Hansen[1], Xiaoli Guo Larsén[1], Maarten Paul van der Laan[1], Pierre-Elouan Réthoré[1], and Juan Pablo Murcia Leon[1]

[1]DTU Wind Energy, Denmark

**Correspondence:** Jana Fischereit (janf@dtu.dk)

**Abstract.** Numerical wind resource modelling across scales from mesoscale to turbine scale is of increasing interest due to the expansion of offshore wind energy. Offshore, wind farm wakes can last several tens kilometres downstream and thus affect the wind resources of a large area. So far, scale-specific models have been developed and it remains unclear, how well the different model types can represent intra-farm wakes, farm-to-farm wakes as well as the wake recovery behind a farm. Thus, in the

present analysis the simulation of a set of wind farm models of different complexity, fidelity, scale and computational costs are compared among each other and with SCADA data. In particular, two mesoscale wind farm parameterizations implemented in the mesoscale Weather Research and Forecasting model (WRF), the Explicit Wake Parameterization (EWP) and the Wind Farm Parameterization (FIT), two different high-resolution RANS simulations using PyWakeEllipSys equipped with an actuator disk model, and three rapid engineering wake models from the PyWake suite are selected. The models are applied to the Nysted

and Rødsand II wind farms, which are located in the Fehmarn Belt in the Baltic Sea.

Based on the performed simulations, we can conclude that average intra-farm variability can be captured reasonable well with WRF+FIT using a resolution of 2 km, a typical resolution of mesoscale models for wind energy applications, while WRF+EWP underestimates wind speed deficits. However, both parameterizations can be used to estimate median wind resource reduction caused by an upstream farm. All considered engineering wake models from the PyWake suite simulate intra-farm wakes

comparable to the high fidelity RANS simulations. However, they considerably underestimate the farm wake effect of an upstream farm although with different magnitudes. Overall, the higher computational costs of PyWakeEllipSys and WRF compared to PyWake pay off in terms of accuracy for situations when farm-to-farm wakes are important.

## 1   Introduction

Numerical wind resource modelling for wind energy covers scales from wind turbine level to meso- and macro-scale of synoptic

systems (Porté-Agel et al., 2020; Veers et al., 2019; Fig. 1). In the past, targeted models have been developed for the different scales. However, with the increasing expansion of wind energy, atmospheric processes that involve different scales are of increasing relevance: individual microscale wind turbine wakes converge to a wind farm wake downstream of a farm. These wind farm wakes can extend more than 50 km both onshore (Lundquist et al., 2019) and offshore (Cañadillas et al., 2020) and thus affect wind resources of neighbouring farms. In addition, they are affected by processes on the mesoscale and synoptic



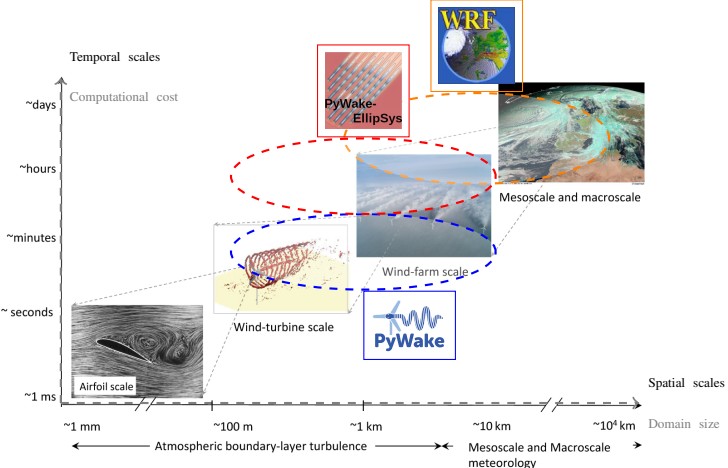

**Figure 1.** Scales relevant for wind energy and targeted models with their associated resolution and computational costs. Taken from Porté-Agel et al. (2020) in accordance with the Creative Commons Attribution (CC BY) license with modifications.

scale, which become increasingly relevant as the studied area become larger (Vincent et al., 2013; Mehrens et al., 2016). Thus, flow over larger areas is not uniform. Rapid engineering models or other high-resolution models that are targeted for the use of microscales around turbines often do not account for this non-uniformity. Instead mesoscale models equipped with a wind farm parameterization (WFP) can be applied to capture these mesoscale variabilites. However, with a typical resolution of about 1–2 km used in wind energy applications according to Fischereit et al. (2021a), mesoscale models cannot resolve individual turbines within a farm.

With this in mind, the aim of this study is three-fold: Firstly, to evaluate the performance of WFPs included in a mesoscale model using a typical horizontal resolution applied in wind energy application against mast measurements and SCADA data and high resolution wake models in terms of intra-farm wakes. Secondly, to compare farm-to-farm wakes and long distance wakes of WFPs and high resolution wake models. Since different model types do not only differ in the typical domain size and therefore in the spatial scales that they can capture, but also by their computational costs (Fig. 1), this study thirdly also addresses the question, whether higher computational costs pay off, when modelling intra-farm and farm-to-farm wakes.

For the investigations, we employ as mesoscale model the Weather Research and Forecasting (WRF) v4.2.2 model (Skamarock et al., 2019) equipped with the Explicit Wake Parameterisation (EWP, Volker et al., 2015) and the wind farm parametrization by Fitch et al. (2012) (FIT). FIT and EWP are the most applied WFPs according to a recent review (Fischereit et al., 2021a). While a few studies have investigated intra-farm wakes using FIT (Jiménez et al., 2015; Eriksson et al., 2015, 2017), only one study (Hansen et al., 2015) evaluated intra-farm wakes with EWP. In addition, in Hansen et al. (2015) WRF was used in a very coarse horizontal resolution so that the entire wind farm was located within one grid cell. In contrast to that Jiménez et al. (2015) and Eriksson et al. (2015) used WRF wihta very high horizontal resolution of 333 m and Eriksson et al. (2017) even a resolution of 111 m. These resolutions are within the terra incognita numerical region or the "grey" zone of the applicability





of mesoscale models (Wyngaard, 2004; Honnert et al., 2020). Within that zone, large coherent overturning structures start to

be partially resolved when their length scale is approaching the effective grid resolution. This violates assumptions that are
applied in the turbulence and shallow convection parameterizations and thus reduces the accuracy of the numerical simulations
in that zone (Honnert et al., 2020). In the present study, a typical resolution for wind energy applications (Fischereit et al.,
2021a), namely 2 km, is applied.

The intra-farm wakes in WRF-EWP and WRF-FIT are evaluated both against SCADA data as well as high resolution
Computational Fluid Dynamics Reynolds-Averaged Navier-Stokes (CFD-RANS) and engineering wake modelling. We employ
the RANS model PyWakeEllipSys (DTU Wind Energy, 2021), which is based on the EllipSys3D CFD flow solver (Michelsen,
1992; Sørensen, 1994) and three different engineering wake models included in the PyWake suite (Pedersen et al., 2019). High-
resolution wake models are typically applied to single wind farms. Only few studies have applied engineering wake models to
farm-to-farm cases (Nygaard and Hansen, 2016; Nygaard et al., 2020; Larsén et al., 2019) or to evaluate long-distance wakes
(Nygaard and Newcombe, 2018). These studies found that simple wake models can in principle represent farm-to-farm wakes
if no coastal gradient is present, but did not compare the performance of different engineering wake models, which will be
done in this study.

The paper is structured as follows. In Sec. 2 the applied mesoscale and high-resolution wake models as well as the available
measurements are introduced. The results are presented in Sec. 3 and split into two parts. In the first part, WRF simulation
results are evaluated against mast measurements (Sec. 3.1). In the second part, the performance of the different wake models to
represent intra-farm (Sec. 3.2) and farm-to-farm wakes (Sec. 3.3) is evaluated for a flow case including farm-to-farm effects. In
addition, the representation of the global blockage effect upstream of a wind farm in the different models is briefly compared
in Sec. 3.4. The results are summarised and discussed in Sec. 4.

## 2 Method

The Fehmarn Belt with the wind farms Nysted and Rødsand II has been selected as the study area (Fig. 2). This area was chosen
since it offers the opportunity to study farm-to-farm effects between Nysted and Rødsand II and mast measurements as well as
supervisory control and data acquisition (SCADA) control system data for Rødsand II wind farm were available for evaluation.
Details on the two wind farms are given in Table 1. Power, thrust and rotational speed curves of the two wind turbine models
are shown in Fig. 3. The curve of the rotor speed is simplified for the present analysis. In reality the Bonus wind turbine shows
a hysteresis behaviour between 5 and 7 $ms^{-1}$: approaching this region from lower wind speeds, the rotor speed is kept at the
minimum value of 11 rpm, while approaching from higher wind speeds the maximum value of 16.5 rpm is kept, as explained
in Nygaard and Hansen (2016). Thus, the linear interpolation between the two rotor speeds between 7 and 8 $ms^{-1}$ as used in
this study, is a simplification. However, since we focus on Rødsand II and not on Nysted for the detailed analysis of internal
wakes, this will not strongly influence the results. In addition, the effect of wake rotation on the velocity deficit is small under
neutral conditions (van der Laan et al., 2015a). More details on the available observations are given in Sec. 2.1; the applied
models and their set-ups are described in Sec. 2.2.

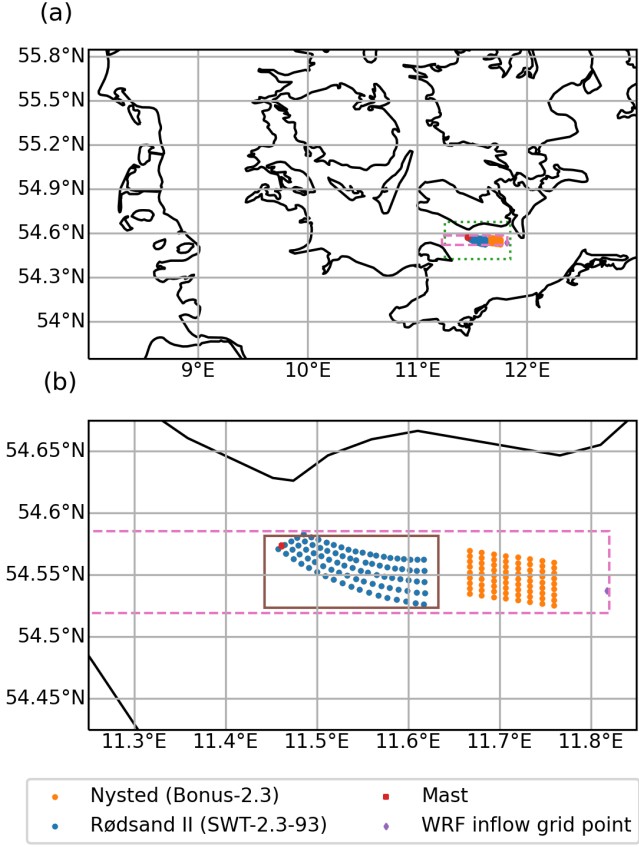

**Figure 2.** Study area 'Fehmarn Belt' with wind farms and mast location. (b) is a zoom of dotted green area in (a). The dot in the east refers to the WRF grid point used for the filtering in Sec. 2.2.3. The solid and dashed rectangle in (b) are used for the analysis of the farm-to-farm wake effect in Fig. 14. Details of the wind farms are given Table 1.

**Table 1.** Characteristics of simulated wind farms in Fig. 2

| Farm | Center lon | Center lat | # Turbines | Model | Hub height [m] | Rotor diameter [m] |
|------|-----------|-----------|-----------|-------|----------------|--------------------|
| **Nysted** | 11.713429 | 54.547488 | 72 | Bonus-2.3 | 69 | 82.4 |
| **Rødsand II** | 11.543771 | 54.556882 | 90 | SWT-2.3-93 | 68.5 | 93 |

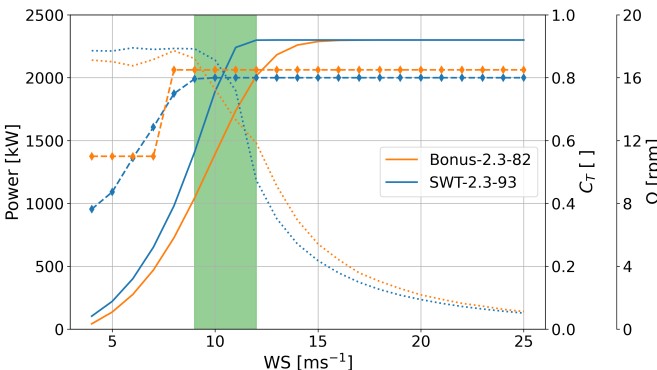

**Figure 3.** Power (left axis, solid lines), thrust curve ($C_T$, right axis, dotted lines) and rotor speed ($\Omega$, rightmost axis, dashed lines with diamonds) for the two turbine models of this study. The shaded green area indicates the investigated wind speed range in this study.

## 2.1 Measurements

Wind measurements with Sonic anemometers at three heights (Table 2) are available from a mast located at the western side of Rødsand II (red dot in Fig. 2). Sonic anemometers also measure air temperature, $T$, which is used to derive the Obukhov length, $L$, based on Lange et al. (2004) in the following way

$$L = \frac{-T \cdot (u_*)^3}{9.81 \cdot 0.4 \cdot \overline{w'T'}} \tag{1}$$

to characterize the stability conditions. Here, $\overline{w'T'}$ is the covariance of vertical wind speed and temperature fluctuations and $u_*$ is the friction velocity, which is derived from the sonic signals. All values are taken from the measurements at 57 m height. Based on the derived value for $L$ stability is classified according to Table 3, which has been taken from Hansen et al. (2012).

In addition, 10-minute SCADA data is available for all 90 turbines of Rødsand II from January 2013 until end of June 2014. The SCADA data include electric power, pitch angle, rotor speed, yaw position and nacelle wind speed. The SCADA data has been quality-controlled (Hansen et al., 2015). From the SCADA data, the equivalent wind turbine wind speed was derived from 10-minute values of power and pitch combined with power and pitch curves. During October 2013, mast measurements were available 88 % of the time and SCADA data were available 100 %. Therefore, this period has been chosen for this study. Since no SCADA data for the turbines of Nysted were available, we assume for this study that Nysted was online and operated with 100% capacity during that period following the discussion in Hansen et al. (2015).

## 2.2 Applied models and set-ups

Three different types of models are employed in this study. The models and set-ups of these are described for the CFD-RANS model from PyWakeEllipSys in Sec. 2.2.1, for the engineering wake model suite PyWake in Sec. 2.2.2 and for the mesoscale model WRF in Sec. 2.2.3, respectively.





**Table 2.** Available instruments and periods for the mast (red dot in Fig. 2) and SCADA data of Rødsand II.

| Instrument | Heights [m amsl] | Period |
|---|---|---|
| Sonic anemometer | 15, 40, 57 | $06.2010 - 06.2015$ |
| SCADA (Rødsand II) | n/a | $01.2013 - 06.2014$ |

**Table 3.** Stability classification based on the Obukhov length, $L$, derived from Eq. 1.

| Stability classification | Range |
|---|---|
| Very stable | $10 \leq L < 50$ |
| Stable | $50 \leq L < 200$ |
| Near stable | $200 \leq L < 500$ |
| Neutral | $500 \leq |L|$ |
| Near unstable | $-500 < L \leq -200$ |
| Unstable | $-200 < L \leq -100$ |
| Very unstable | $-100 < L \leq -50$ |

### 2.2.1 CFD-RANS wake modelling using PyWakeEllipSys

We apply the RANS model from PyWakeEllipSys v1.5.1 (DTU Wind Energy, 2021), which is based on the EllipSys3D CFD flow solver initially developed by (Michelsen, 1992; Sørensen, 1994). EllipSys3D is an incompressible finite volume flow solver using a block-structured grid. PyWakeEllipSys is developed to simulate the wind turbine interaction subjected to an atmospheric inflow that can represent effects of turbulence intensity, atmospheric stability, ABL height and Coriolis forces (van der Laan et al., 2021a). The inflow models are all based on modified $k$-$\varepsilon$ turbulence model closures.

The numerical domain represents an o-grid of about 160 km in diameter and has two nested refined inner regions, as depicted in Fig. 4. The most inner region (blue dashed box in Fig. 4 has uniformly spacing in the horizontal directions using a cell size of $D/8$, based on a grid refinement study from van der Laan et al. (2015b), where $D$ is the rotor diameter of the Bonus-2.3 wind turbine. The outer refinement region (magenta dashed box in Fig. 4 is used to resolve the wind farm wake using a maximum horizontal spacing of $1D$. The first cell height is set to 1.5 m and grows initially to $D/8$ at a height of $3D$, and then continues to grow up until a domain height of $25D$ is reached. In total 258 million cells are used. A rough wall boundary condition (Sørensen et al., 2007) is set as the ground and an inflow condition is used at the top of the domain. The lateral boundaries are either inflow or outflow boundaries depending on the inflow wind direction. The outflow boundary has a angle of $45°$, at which all gradients normal to the boundary are assumed to be zero.

We have chosen to employ two inflow models; a pure neutral inflow following a logarithmic profile and a near-neutral inflow including an ABL height and Coriolis forces; the models are referred to as $k$-$\varepsilon$ and $k$-$\varepsilon$ ABLc in van der Laan et al. (2021a), respectively, where more information can found. In the present work, we use RANS-ASL and RANS-ABL to distinguish

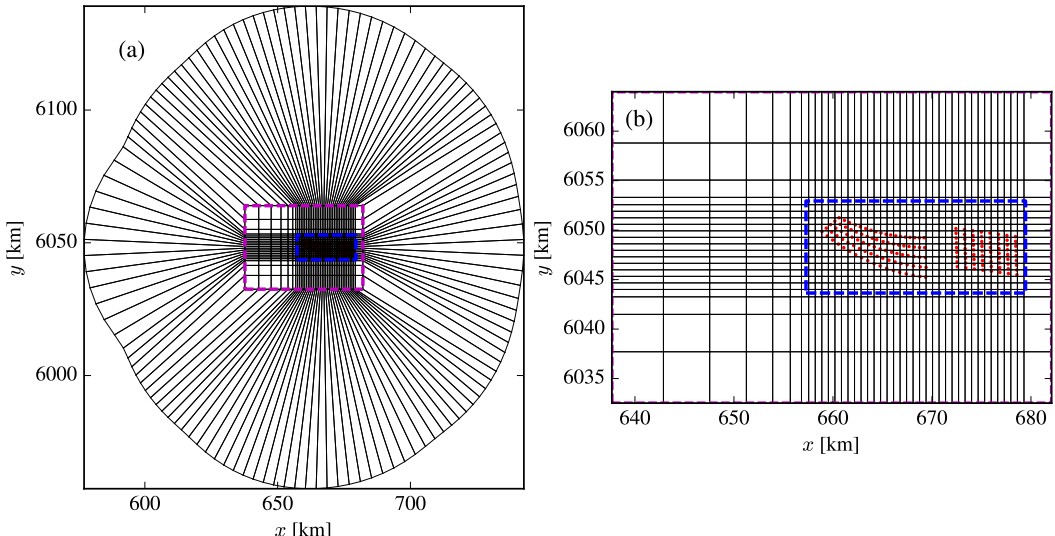

**Figure 4.** Surface grid of RANS simulations where every 64th grid line is shown. Right plot **(b)** is a zoomed view of left plot **(a)**. Two inner regions are marked by the magenta and blue dashes boxes. Wind farm layouts are shown as red dots.

the $k$-$\varepsilon$ and $k$-$\varepsilon$ ABLc models, respectively. The employed inflow profiles are depicted in Fig. 5. The RANS-ASL model represents a neutral surface layer and only has the roughness length as input parameter, which we set as $9.33 \times 10^{-4}$ m to get a TI of 7% at hub height. The friction velocity is used the scale the inflow profile to get the desired hub height wind speed, and its value is found by a one-dimensional precursor simulation that includes the same numerical errors close to the wall as the 3D wind farm simulation, although the shape of the resulting profile is very similar to the analytical logarithmic profile. The use of one-dimensional inflow precursor assures that the inflow is in full balance with the three-dimensional domain, which is a requirement to investigate small effects as wind farm blockage. The RANS-ABL model represents an ABL height by setting a maximum turbulence length scale, $\ell_{\max}$ following Apsley and Castro (1997). For small values of $\ell_{\max}$, a shallow ABL height is obtained and the resulting ABL profiles resemble stable conditions, without the need for a temperature equation. We choose a roughness length of $5 \times 10^{-3}$ m and set the Coriolis parameter to $1.187 \times 10^{-4}$ s$^{-1}$ corresponding to a latitude of 54.5 °. A pre-calculated ABL library using Rossby similarity (van der Laan et al., 2021b) is used to find the geostrophic wind speed $G$ and $\ell_{\max}$ that sets the desired turbulence intensity and wind speed at the reference height of 69 m. Four wind speeds at hub height are employed ($WS_{69}$), namely 9, 10, 11 and 12 m/s (see Sec. 2.3.1), and we find $G = 11.3, 12.8, 14.3, 15.8$ m/s and $\ell_{\max} = 38.3, 34.9, 32.3, 30.3$ m, respectively. The obtained values of $\ell_{\max}$ can be related to the Obukhov length as $L \approx \ell_{\max}/0.08$ following Apsley and Castro (1997), which gives a range of $L \approx 380 - 480$ m, resembling near stable conditions as defined by Table 3. Figure 5f shows that the wind veer over the rotor area is about 2°.



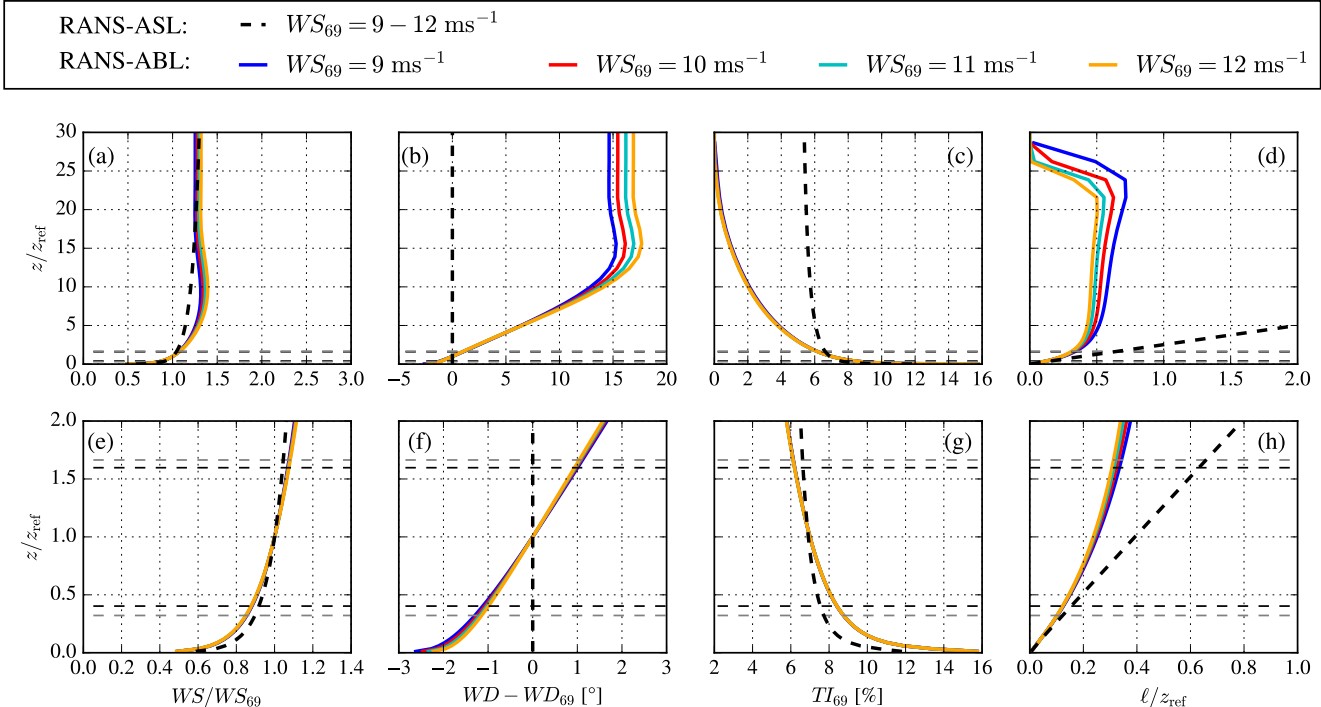

**Figure 5.** Inflow profiles for RANS simulations. Bottom plots **(e-h)** are a zoomed view of the top plots **(a-d)**, focused around wind turbine rotor areas, which are depicted as black (Bonus-2.3) and gray (SWT-2.3-93) dashed lines. **(a, e)** Wind speed (WS). **(b, f)** Wind direction (WD). **(c, g)** Turbulence intensity (TI). **(d, h)** Turbulence length scale ($\ell$).

The standard $k$-$\varepsilon$ turbulence model underpredicts the wake deficits due to an overestimation of the near wake turbulence length scale (van der Laan and Andersen, 2018). The issue has been mitigated by the addition of a local turbulence length scale limiter, $f_P$. The resulting $k$-$\varepsilon$-$f_P$ model has been developed and validated for wind farm RANS simulations under neutral surface layer conditions (van der Laan et al., 2015b,a). In this work, we use the $k$-$\varepsilon$-$f_P$ model for the RANS-ASL setup and

135 we also couple the $f_P$ function with the $k$-$\varepsilon$-ABL model for the RANS-ABL setup following van der Laan et al. (2021a). The coupling with the RANS-ABL requires more validation and may need to be revised.

PyWakeEllipSys employs an actuator disk model to represent wind turbine forces (Réthoré et al., 2014) and has several methods to model the force distributions. We have chosen the analytic force distribution model from Sørensen et al. (2019) including tangential forces, and effects of shear and veer that result in force distribution variations in the azimuthal direction.

Each actuator disk uses a polar grid with 32 cells in both directions.





### 2.2.2 Engineering wake models

Three different rapid engineering wake models out of the open source wind farm simulation tool PyWake v2.2.0 (Pedersen et al., 2019) are applied in this study. The three wake deficit models, NOJ (Jensen, 1983), BAS (Bastankhah and Porté-Agel, 2014) and ZON (Zong and Porté-Agel, 2020), differ in complexity: The NOJ model has been developed for the far wake and

145 represents the wake as a simple top-hat wake and a linear wake expansion constant of k=0.1. In contrast, BAS assumes a Gaussian distribution for the velocity deficit in the wake and has been derived assuming conservation of mass and momentum. It is valid only for the far wake, i.e. the region starting approximately 2-4 rotor diameters downstream of the turbine when the rotor geometry does not play a dominant role anymore, and has been validated against wind-tunnel measurements and large-eddy simulations with a linear wake expansion constant of k=0.0324555. The last model, ZON, simulates the wake expansion

as function of local TI (with parameters as described in (Zong and Porté-Agel, 2020)) and the approach by Shapiro et al. (2018) is used for the wake width expansion. All three wake models are combined with the same blockage deficit model, SSD, which is an update of the self-similar deficit model by Troldborg and Meyer Forsting (2017). In this update two changes are made, which are documented in Pedersen et al. (2019). First, a linear fit is used radially to avoid large lateral induction tails and second, the axial induction depends now on $C_T$ and axial coordinate. To account for the up- and downstream effect of each turbine,

an iterative approach is chosen to represent the deficit caused by all wind turbines on all wind turbines ("All2Alliterative"). Individual deficits are superimposed using a squared sum for NOJ and BAS or using a linear sum for ZON. More details on the methods are available online (Pedersen et al., 2019).

  The simulations extend over the inner domain of the PyWakeEllipSys simulations (Fig. 4 magenta dashed area), but use 750 equally spaced grid points in east-west direction and 375 equally spaced grid points in north-south direction. This corresponds

to a resolution of 59 m in east-west direction and 84 m in north-south direction.

### 2.2.3 Mesoscale wake modelling using WRF

Mesoscale model simulations for different wind farm scenarios have been conducted with WRF v4.2.2 with the settings in Table 4. WRF simulations are performed using three nested domains with 18 km, 6 km and 2 km horizontal resolution, respectively (Fig. 6). In the vertical a non-equidistant grid is used with a resolution of about 10 m within the rotor area following the recommendations by Siedersleben et al. (2020); Lee and Lundquist (2017); Tomaszewski and Lundquist (2020)

following the recommendations by Siedersleben et al. (2020); Lee and Lundquist (2017); Tomaszewski and Lundquist (2020) and in total 14 levels below 250 m (Fig. 6b).

  Wind farms effects are parameterized using both FIT (Fitch et al., 2012) and EWP (Volker et al., 2015). FIT and EWP differ mainly in two aspects. First, EWP accounts for a sub-grid scale vertical wake expansion based on the concept from Tennekes and Lumley (1972), while FIT does not include sub-grid scale effects. Second, only FIT considers wind farms as

explicit source of Turbulent Kinetic Energy (TKE), while EWP assumes that TKE develops solely through shear production in the wind farm wake. More details on their comparison can be found in (Fischereit et al., 2021a). FIT is applied here including the bug fix provided by Archer et al. (2020) with the recommended TKE coefficient of 0.25. Both parameterizations require



**Table 4.** WRF parameterisation, boundary conditions and forcing data employed for the performed simulations. The wind farm parametrizations EWP and FIT were only applied for some scenarios (Table 7).

| Category | Subcategory | Details (option number) |
|---|---|---|
| WRF | Version | 4.2.2 |
| Time | Simulation length | 5.5 days including 12 h spin-up |
| | Time step | 45 s (30 s for period 25.10.2013 12:00 – 30.10.2013) |
| | Output time step | 10 min |
| Schemes | PBL | MYNN (5) |
| | surface layer | MO (2) |
| | Microphysics | New Thompson et al. scheme (8) |
| | Radiation | RRTMG scheme (4) |
| | Cumulus parameterisation | Kain-Fritsch scheme (1) on domain 1 |
| | Diffusion | Simple diffusion (1)<br>2D deformation (4)<br>6th order positive definite numerical diffusion (2) rates of 0.06, 0.08 and 0.1 for domain 1, domain 2 and domain 3 vertical damping. |
| | Advection | Positive definite advection of moisture and scalars (1)<br>TKE advection turned on |
| | Wind farm parameterisation | EWP (R0frac = 1.7) |
| | | FIT (TKE factor = 0.25) |
| Boundary and forcing data | Dynamical forcing | ERA5 at pressure levels every 6 hours |
| | Land use data | CORINE |
| | Sea surface temperature | OSTIA |
| | Land surface model | NOAH-LSM (2) |

turbine positions, turbine model and thrust and power curves as input. Details on the turbine models are given in Table 1. The power and thrust curve of the two turbine models are visualised in Fig. 3.





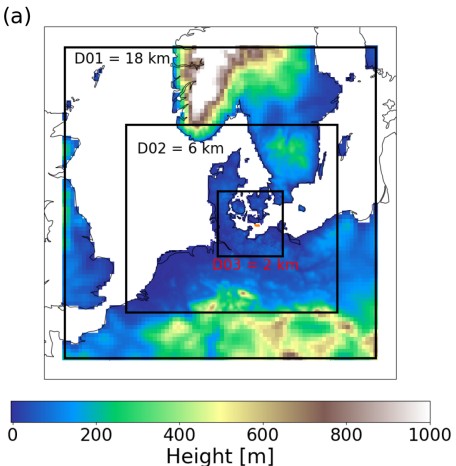

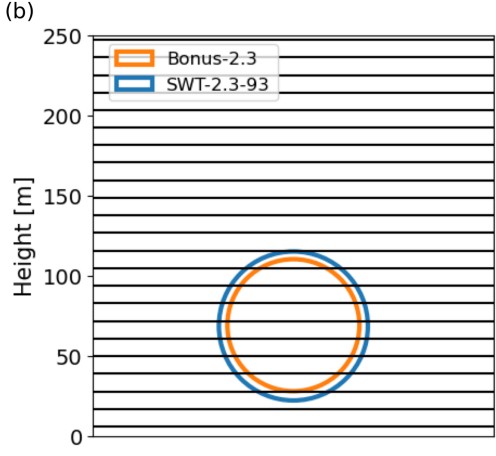

**Figure 6.** (a) Nested WRF domains with the innermost domain capturing the study area of the Fehmarn Belt with the Nysted and Rødsand II wind farms (orange dots) and (b) vertical WRF grid with the lowest 14 mass levels and rotor areas covered by the two wind turbine models.

## 2.3 Simulated scenarios

For WRF a one-month long period, October 2013, is simulated. This period has been selected based on high availability of mast measurements and Rødsand II SCADA data (Sec. 2.1). The simulations are conducted as consecutive 5.5 days periods with an overlap of 12 hours that is disregarded as spin-up. This method, which is similar to the method applied for the New European Wind Atlas (Dörenkämper et al., 2020), has the advantage that model drifts are avoided due to the frequent new initialisation but at the same time unnecessary computations during spin-up are kept to a minimum.

In contrast to WRF, for PyWake and PyWakeEllipSys the simulations do not represent the conditions in October 2013. Instead a farm-to-farm flow case for eastern wind directions is simulated to be able to study the wind farm wake effect of Nysted on Rødsand II and compare the results against the SCADA data available only for Rødsand II (Sec. 2.1). To compare these flow case simulations with the real WRF simulations and real SCADA data, the WRF results and SCADA data are filtered for the same conditions as the simulated flow case. The details of the filtering are described in Sec. 2.3.2.

### 2.3.1 Selected flow case

The impact of wakes on power is largest just below rated wind speed (Lundquist et al., 2019). Thus, a wind speed range around $10 \, \text{ms}^{-1}$ has been selected for this study, which is just below rated power for the SWT2.3-93 turbine of Rødsand II (Fig. 3).

To realize these conditions in PyWakeEllipSys, ten different wind directions between $62.5° - 112.5°$ with a $5°$ interval and four different wind speeds between 9 and $12 \, \text{ms}^{-1}$ as shown in Table 5 are simulated for both inflow models (RANS-ABL and RANS-ASL, Sec. 2.2.1). As a post processing step, the flow variables are Gaussian averaged over the wind directions with a standard deviation of $5°$ and the resulting four wind directions (82.5, 87.5, 92.5 and $97.5°$) are subsequently linearly averaged



over the different wind directions. Finally, the simulations are averaged over the four wind speeds weighted according to the simulated inflow wind speed from the filtered WRF-NWF simulation (Sec. 2.3.2).

The set-up of the simulated flow cases for PyWake is similar to the set-up of PyWakeEllipSys. For PyWake 360 simulations with a 1° interval and four different wind speeds between 9 and 12 ms$^{-1}$ as shown in Table 5 are simulated. As for PyWakeEllipSys, a Gaussian filter for wind direction is applied to the PyWake results with a standard deviation of 5°. Afterwards the results are linearly averaged over averaged over the wind directions between 82° and 98°. The results are finally averaged over the four wind speeds weighted according to the simulated inflow wind speed from the filtered WRF-NWF simulation

(Sec. 2.3.2).

To derive both the intra- and farm-to-farm effect, two different wind farm configurations are simulated with PyWake and PyWakeEllipSys (Table 5). In the first configuration, named NYRØ, the actual situation is simulated with both Nysted and Rødsand II present. In the second configuration, named RØ, only the Rødsand II wind farm is included. Using RØ in combination with NYRØ allows to isolate the effect of Nysted on Rødsand II for the different models.

**Table 5.** Simulated flow cases with the high-resolution wake models in terms of wind speed (WS), wind direction (WD) and inflow turbulence intensity (TI) at 69 m height. For scenario and wake model description see Sec. 2.2.1 and Sec. 2.2.2. Note that RANS-ABL simulations are only approximately neutral, as depicted by the star (*), for details see Sec. 2.2.1. Linear is abbreviated lin.

| Model | Name | Stability | $TI_{69}$ [%] | $WS_{69}$ [ms$^{-1}$] | $WD_{69}$ [°] | Scenarios | Wake | Block-age | Super-pos. | Wind farm |
|---|---|---|---|---|---|---|---|---|---|---|
| PyWakeEllipSys | RANS-ABL | neutral* | | | [62.5, 67.5,...,112.5] lin ave [82.5,97.5] | | | Actuator disc | | |
| | RANS-ASL | neutral | 7 | 9,10, 11,12 | | RØ, NYRØ | NOJ | SSD | Squared sum | All2All-iterative |
| PyWake | NOJ | | | | [0,1,...,360] lin ave [82,98] | | BAS | | | |
| | BAS | neutral | | | | | ZON | | Lin sum | |
| | ZON | | | | | | | | | |

### 2.3.2   Filtering methods

To make the one month long WRF simulations comparable to the flow case simulated by PyWake and PyWakeEllipSys, WRF results for October 2013 have to be filtered for neutral conditions, eastern wind direction and wind speeds around 10 ms$^{-1}$.

Neutral conditions are detected using the Obukhov length as described in Sec. 2.1. For filtering to the wind speed range of interest (9 ms$^{-1}$ – 12 ms$^{-1}$), the wind speed measurements at the Rødsand II mast cannot be used, since they are affected by

the wake of Rødsand II and Nysted and thus do not represent free stream velocities. Instead, the wind speed filtering is based on the WRF simulation without wind farms (NWF, Table 7) for which an upstream grid point from Nysted (Fig. 2) is used to detect the inflow conditions. Due to infrequent joint occurrence of easterly wind with flow in the wind speed range of interest during October 2013 (Fig. 7a), only very few 10-min samples could be identified. For filtering the wind direction, two different



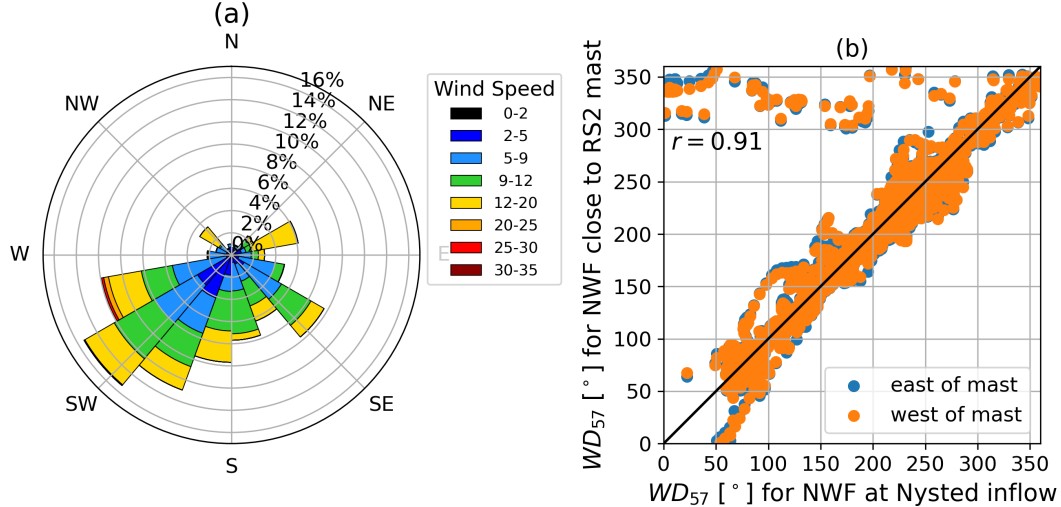

**Figure 7.** (a) Wake affected measured wind rose at the Rødsand II mast at 57 m height for October 2013 and (b) for NWF inflow wind direction at Nysted inflow grid point versus wind direction close to the mast. For locations see Fig. 2.

approaches, named f1 and f2, respectively, have been tested (Table 6). In f1 the wind direction filter is based on the WRF-NWF
simulation at the WRF inflow grid point (Fig. 2). In f2 the filtering is based on observed wind direction at the Rødsand II mast,
which represents more closely the conditions represented by the SCADA data of Rødsand II (Sec. 2.1). Two different methods
are applied, since the WRF-NWF simulations indicated a wind direction change from the inflow to Nysted to the outflow of
Rødsand II (Fig. 7b).

The described filtering for stability, wind speed and wind direction is applied to both SCADA and the WRF simulations
to identify 10-minute periods, which correspond to the simulated flow cases. The identified 10-minute periods are linearly
averaged (Table 6). The flow scenarios simulated by PyWake and PyWakeEllipSys for the four inflow wind speeds 9, 10, 11
and 12 ms$^{-1}$ are averaged weighted according to the WRF-NWF wind speed at the WRF inflow grid point (Fig. 2).

**Table 6.** Details on filter method 1 and 2 and corresponding available number of 10-minute period, average inflow wind direction ($WD$) and average inflow wind speed ($WS$). The subscript "in" refers to the WRF inflow grid point, which is depicted along with the other locations in Fig. 2.

| Method | Wind direction filter | Number of periods | $\overline{WD}_{69,in}$ [°] | $\overline{WS}_{69,in}$ [ms$^{-1}$] |
|--------|----------------------|-------------------|------------------------------|--------------------------------------|
| f1 | $80 \leq WD_{60,in} \leq 100°$ for NWF at inflow grid point | 7 | 93 | 10.2 |
| f2 | $80 \leq WD_{57} \leq 100°$ for mast at Rødsand II | 19 | 112 | 9.7 |





As for PyWake and PyWakeEllipSys, for WRF also NYRØ and RØ wind farm scenarios are simulated, which include Nysted and Rødsand II and Rødsand II only, respectively. In addition a third simulation, NWF, without any wind farm is performed.

**Table 7.** Periods and scenarios covered by the WRF simulations.

| Period | Parameterised wind farm | WFP | Name |
|---|---|---|---|
| | No | | NWF |
| 01 October 2013 – 30 October 2013 | Rødsand II | EWP (Volker et al., 2015) | RØ-EWP |
| | | FIT (Fitch et al., 2012; Archer et al., 2020) | RØ-FIT |
| | Nysted, Rødsand II | EWP (Volker et al., 2015) | NYRØ-EWP |
| | | FIT (Fitch et al., 2012; Archer et al., 2020) | NYRØ-FIT |

## 3 Results

The analysis of this study is split into two parts. First, the full one-month long WRF simulations are evaluated against mast measurements in Sec. 3.1. Second, the filtered WRF simulations and SCADA data are used to evaluate and compare the simulation results of the different wake models for intra-farm wakes (Sec. 3.2), farm-to-farm wakes (Sec. 3.3) and global blockage (Sec. 3.4).

### 3.1 Evaluation of WRF with mast measurements

The WRF simulations are compared against sonic measurements at the Rødsand II mast for October 2013 in Fig. 8 for both wind speed ($WS$) and wind direction ($WD$). In general, the time series for both $WS$ and $WD$ agree well with the sonic measurements. One exception is the peak in $WS$ during storm Christian on 28 October, which is not fully captured by WRF. However, during such high wind speeds, the wind turbines of Rødsand II and Nysted are turned off (Fig. 3) and thus this miss does not impact the evaluation. The good visual agreement is confirmed by the high correlation coefficient of 0.88 for $WS$ and 0.91 for $WD$ (Fig. 9).

The simulation results gradually improve from NWF to NYRØ-EWP and NYRØ-FIT due to accounting for the wind farm effect (Fig. 9). The $WS$-bias reduces from 1.04 ms$^{-1}$ for NWF to 0.8 ms$^{-1}$ for EWP and 0.68 for FIT (Fig. 9a,b,c). The $WD$-bias reduces from 8.45° for NWF to 7.76° for EWP and 7.74° for FIT. Thus, FIT clearly performs best for simulating the wind deficit, the improvements for $WD$ compared to NWF are similar for EWP and FIT. Looking at the black binned line in Fig. 9a,b,c, representing mean and standard deviations, it becomes clear that compared to NWF both EWP and FIT show the largest improvements around medium high wind speeds, where the thrust is high (Fig. 3).



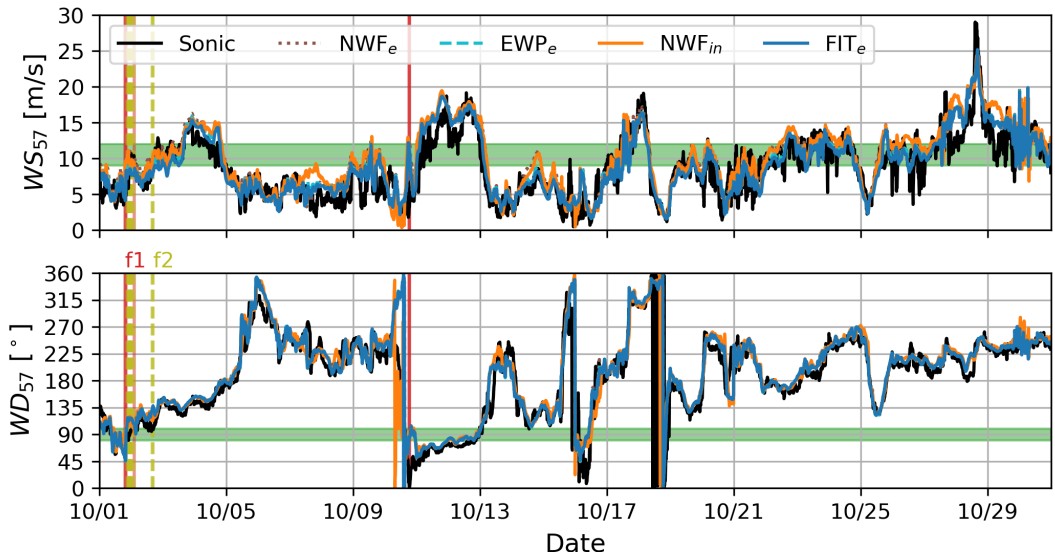

**Figure 8.** Time series of wind speed ($WS$, top) and wind direction ($WD$, bottom) at 57 m height for October 2013 for sonic mast measurements (black), and for the closest WRF grid point to the east (subscript $e$) of the mast for NWF (dotted brown), NYRØ-EWP (dashed light blue), NYRØ-FIT (solid blue) and NWF at the inflow grid point, NWF$_{in}$ (Fig. 2). The horizontal patch marks the ranges of interest and vertical lines the two filter methods $f1$ (solid red) and $f2$ (dashed yellow). See Table 6 and Table 7 for the abbreviations.

The wind farm wake effect for the entire October 2013 at the mast can be derived by subtracting the $WS$-bias of NWF from the $WS$-bias of EWP or FIT, respectively. Doing so, indicates a wake effect at 57 m height, i.e. 12 m below hub height, of
0.2 ms$^{-1}$ for EWP and 0.36 ms$^{-1}$ for FIT. These rather small wake effects can be explained by the location of the mast at the west of Rødsand II (Fig. 2), and the relative frequent wind direction for October 2013 from south-west (Fig. 8), when the mast is not much influenced by the wake. Performing the same analysis filtered for eastern wind directions ($90° \pm 10°$) using the measured wind direction at the mast (f2 in Table 6) indicates a considerable larger wake effect of 0.6 ms$^{-1}$ for EWP and 1 ms$^{-1}$ for FIT at 57 m height based on 127 10-minute values.



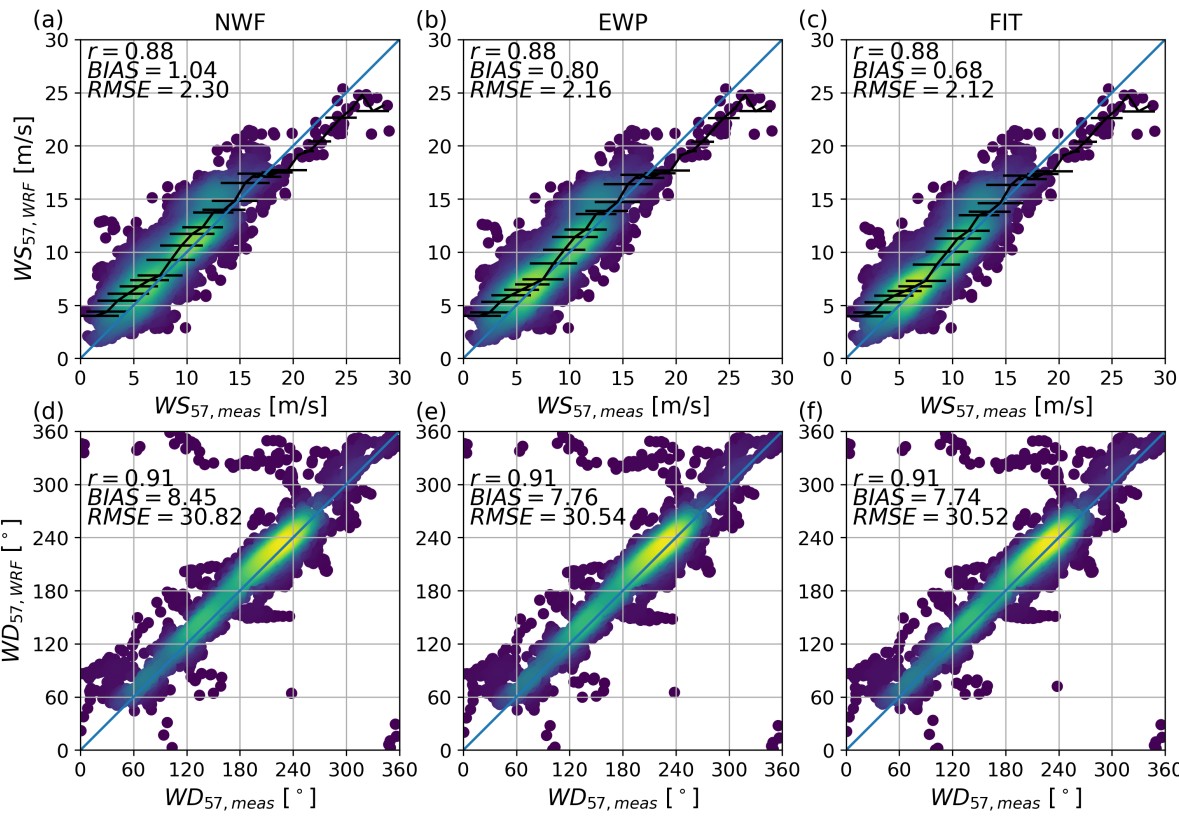

**Figure 9.** Scatter plot of measured (a),(b),(c) wind speed ($WS$) and (d),(e),(f) wind direction ($WD$) in 57 m height against WRF simulations namely (a),(d) NWF, (b),(e) NYRØ-EWP and (c),(f) NYRØ-FIT (Table 7) for October 2013.

## 3.2 Comparison of intra-farm wake modelling

### 3.2.1 Comparison of filter methods

To compare the flow cases modelled with PyWakeEllipSys and Pywake against the real WRF simulation and SCADA data for October, two different filter methods were proposed (Table 6). The two methods are compared in Fig. 10 for different south-north transects at different locations within the area of the wind farms in east-west direction with (a) being the eastern-most transect and (f) being the western-most transect. Normalized relative coordinates are used by transforming the original coordinates $x$ using

$$x_{rel,n} = \frac{x - x_{ref,t}}{D} \tag{2}$$

where $x_{ref,t}$ is the location of the eastern most turbine of Rødsand II and $D$ is the rotor diameter of the Rødsand II turbine model SWT-2.3-93 (Table 1). Only the simulation results for WRF-NWF, WRF-FIT and RANS-ABL are shown. Solid lines



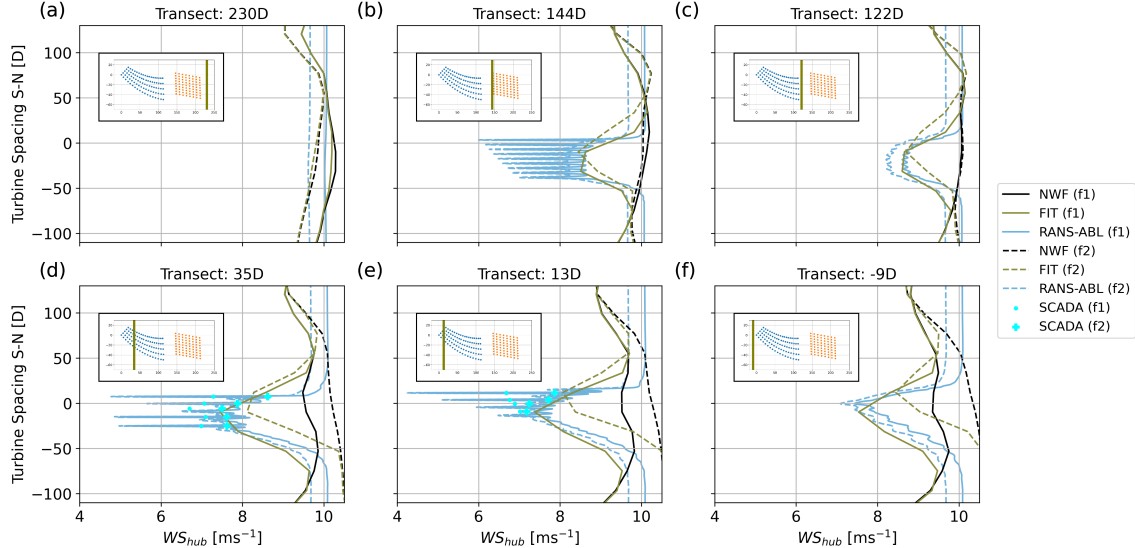

**Figure 10.** North-south transects for wind speed at hub height for different models at different locations in east-west direction as indicated by the olive line in each subplot and the transect name in the title in normalized coordinates as explained in the text. All simulations are for NYRØ. For abbreviations see Table 7 and Table 5. Solid lines and dots indicate results for the f1 method, dashed liens and pluses indicate results for the f2 method (Table 6).

indicate results according to f1 whereas dashed lines refer to f2. Equivalent wind speed derived from SCADA data for the turbines close to the transect and filtered according to f1 and f2 are shown as dots and pluses, respectively.

The WRF-NWF simulations show that while the two filtering methods agree well for the transects 144D and 122D between Nysted and Rødsand II, they disagree for the inflow transect to Nysted (230D) and especially for the southern part of the transects in the area of Rødsand II (35D, 13D, -9D). In accordance with the smaller average wind speed in f2 (Table 6), all f2 results are shifted towards smaller values. The simulations also reflect the average inflow directions for f1 and f2 (Table 6) in that the maximal wind speed deficits for f2 are shifted north-wards compared to f1 and the RANS-ABL results.

In contrast, SCADA data from f2 agree better with RANS-ABL compared to f1. This is because for f2 the filtering uses the wind direction measurements at the Rødsand II mast, which is closer to Rødsand II than the inflow to Nysted for f1. The analysis of wind directions to the east and to west of the two farms indicate for WRF-NWF indicate wind direction changes over the domain (Fig. 7b). This could be related to coastal effects from Lolland but deserves further analysis. However, it indicates that even in this relatively small area of about 20 km, wind directions cannot simply be assumed to be unified across the area. This indicates that flow case simulations, like those performed with PyWake and PyWakeEllipSys in this study, need to be taken with care as the area of interested increases and when coastal effects could play a role.

The comparison of f1 and f2 indicates that both methods have advantages and disadvantages: f1 represents better the inflow conditions of the simulated flow cases for PyWake and PyWakeEllipSys, while f2 compares better with the rotor equivalent wind speed from SCADA. Thus, depending on the target analysis either method is used in the following.





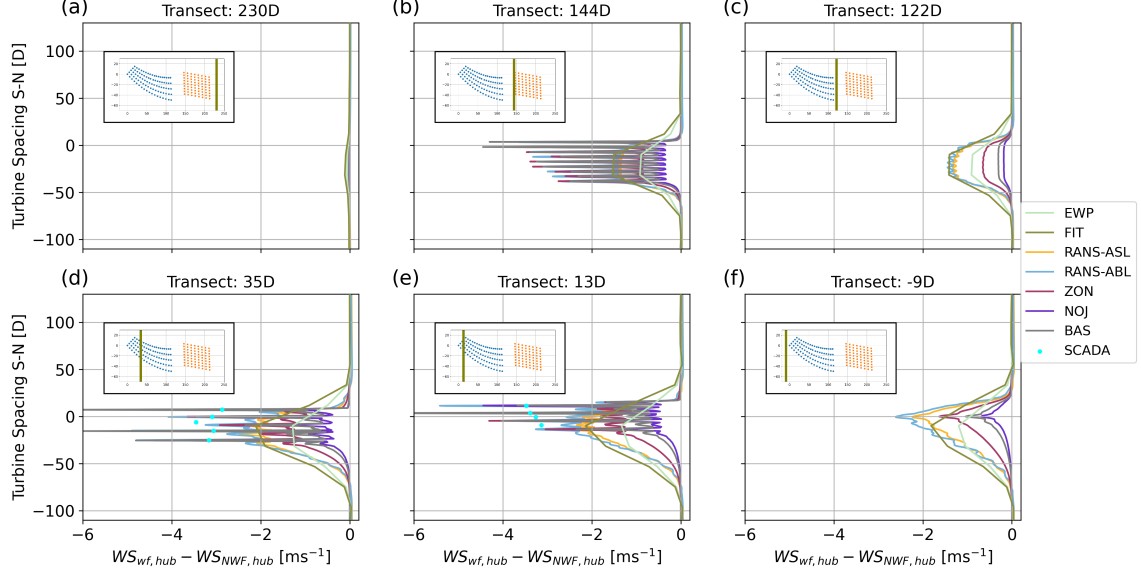

**Figure 11.** Wind speed transects at hub height for the different models in north-south direction at different locations as indicated by the olive line in each subplot. The coordinates are relative to the coordinates of the eastern most turbine of Rødsand II and normalized with the rotor diameter of the Rødsand II turbine model SWT-2.3-93 (Table 1). All simulations are for NYRØ and normalized with a no-wind-farm (NWF) scenario. For abbreviations see Table 7 and Table 5. WRF results and SCADA data are filtered according to the f1 method (Table 6).

### 3.2.2 Comparison of models

To facilitate the comparison between the real WRF simulations and the flow cases simulated with the other models as described in Sec. 2.3.1, the results are normalized with a no wind farm scenario. For WRF-FIT and WRF-EWP, WRF-NWF is used to

do so. For the PyWake and PyWakeEllipSys simulations, the eastern most wind speed is used, which is not influenced by the presence of the wind farms. The normalized results are shown for the same transects as in Fig. 10 for the two filter filter methods in Fig. 11 and Fig. 12, respectively. The figures show the simulation results of all applied models abbreviated in accordance with Table 5 and Table 7.

    For both filter methods, the transect 230D (Fig. 11a, Fig. 12a) show a global blockage effect of Nysted for the WRF and

PyWakeEllipSys simulations. This will be discussed further in Sec. 3.4.

    Transects 144D, 35D and 13D (Fig. 11b,d,e Fig. 12b,d,e, respectively) are taken within or just outside the farms. As expected due to the coarse resolution WRF-EWP and WRF-FIT cannot resolve the internal variability within the farms with large deficits of up to 5 ms$^{-1}$ very close to the individual turbines. However, especially WRF-FIT compares very well with the average deficits of RANS-ASL and RANS-ABL for all transects for f1 (Fig. 11). This is also true for f2 (Fig. 12), except that the

maximum deficit is shifted north-ward in WRF due to the average wind direction being more southerly (Table 6) as discussed in Sec. 3.2.1. In contrast to FIT, EWP underestimates wind speed deficits when compared with both RANS-ASL and RANS-





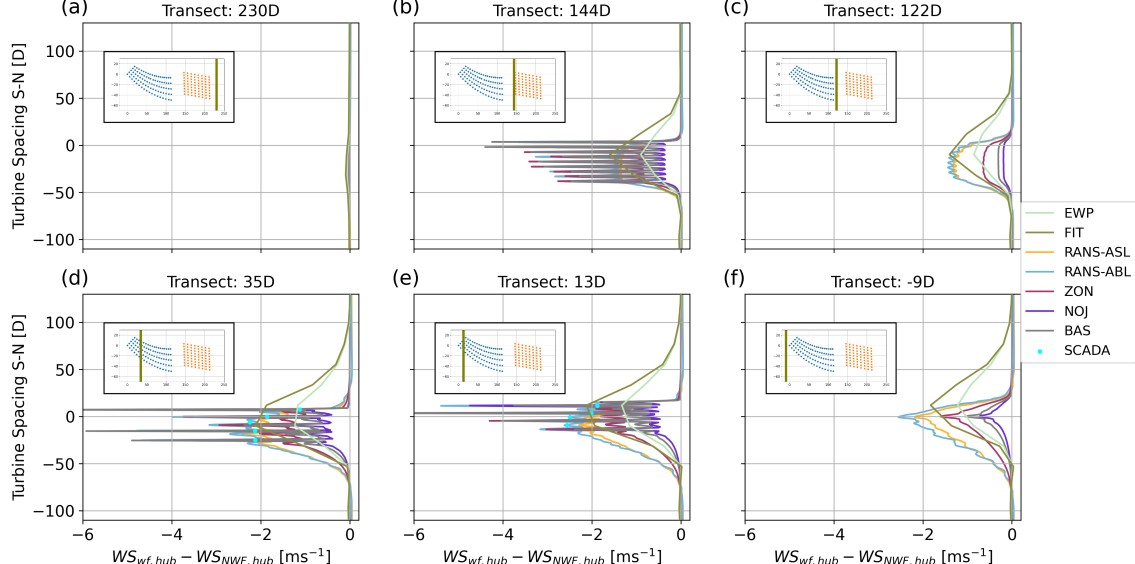

**Figure 12.** As Fig. 11 but for filter method f2.

ABL. The equivalent wind speeds derived from SCADA data indicate that both RANS and WRF-FIT results are reasonable. This is true especially for f2 (Fig. 12), which better represents the actual situation close to Rødsand II as discussed in Sec. 3.2.1.

The results for the two RANS methods (Sec. 2.2.1) give pretty similar results. This is because the RANS-ABL model is 295 employed with a relative large ABL height resembling near-neutral conditions and the resulting inflow can be approximated by an atmospheric surface layer, as used by the RANS-ASL model.

The peak deficits of all applied wake models in PyWake, ZON, NOJ and BAS, agree relatively well with the peak deficits for PyWakeEllipSys (transects 144D, 35D and 13D). However, the average deficits in PyWake are much smaller and are even smaller than WRF-EWP. In addition the deficits are also narrower in south-north direction around the borders of the farm 300 compared to the RANS results, especially for the transects downwind for Nysted (Fig. 11c-f). Downstream of the farms, the flow recovery for PyWake is very rapid, except for ZON. Thus, the inflow to Rødsand II is not much reduced for NOJ and BAS compared to the free-stream inflow to Nysted. This indicates that the chosen wake models in PyWake can represent the wind speed deficits close to turbines within one wind farm well, but underestimate the effect of farm-to-farm wakes with different magnitudes as will be discussed in more detail in Sec. 3.3.

**3.3 Comparison of farm-to-farm wake modelling**

To isolate the effect of Nysted on Rødsand II, the difference between NYRØ and RØ, i.e. a simulation without Nysted, is investigated. Here only the results for f1 are presented, since the inflow conditions for those simulations agree better with the inflow conditions of the PyWake and PyWakeEllipSys simulations than the results for f2.



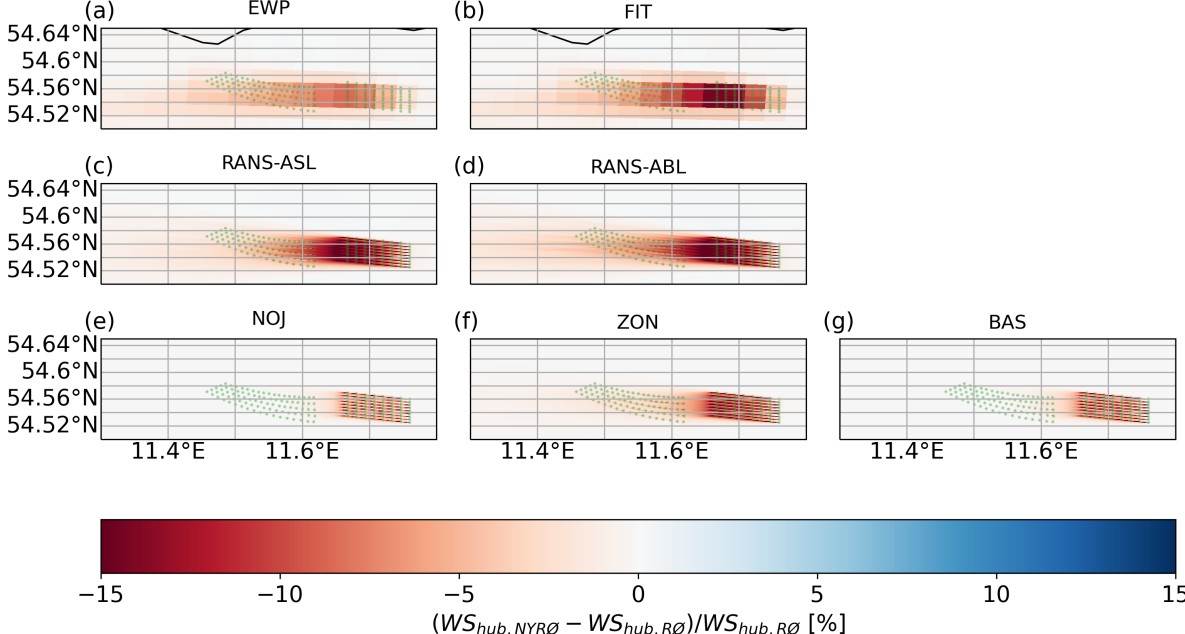

**Figure 13.** Relative difference in hub height wind speed ($WS$) between NYRØ and RØ different wake models for filter method f1. For abbreviations see Table 5 and Table 7.

Fig. 13 shows spatially the relative wind speed difference at hub height between NYRØ and RØ for the different models.
As expected from the intra-farm analysis in Sec. 3.2.2, WRF-FIT results agree very well with the high-fidelity RANS results (Fig. 13b and Fig. 13c,d). WRF-EWP underestimates the wind speed reduction due to the presence of Nysted compared to RANS. While PyWake results (Fig. 13e,f,g) show strong deficits behind individual turbines, the wakes are very narrow and the influence on Rødsand II is much smaller compared to the results of the other models, as expected from the analysis in Sec. 3.2.2 already. Out of the three investigated deficit models in PyWake, ZON shows the strongest farm-to-farm effect, which
is, however, still smaller than the influence based on RANS and WRF.

The impact of Nysted on the flow extends beyond Rødsand II for EWP, FIT, RANS-ASL and RANS-ABL as indicated by reduced wind speeds west of Rødsand II. Comparing RANS-ASL and RANS-ABL, RANS-ABL shows slightly longer and stronger wakes. This is expected, since simulated inflow profile corresponds to slightly stable conditions, where wakes are generally longer (Cañadillas et al., 2020).
To quantify the impact of Nysted on Rødsand II, the wind speed and power reduction in the area of Rødsand II is shown in from of histograms in Fig. 14. In all sub-plots the difference between NYRØ and RØ are shown for different deficit classes as simulated over brown area in Fig. 2. Fig. 14a,c show absolute differences for wind speed and power, respectively, between NYRØ and RØ and Fig. 14b,d normalized difference with the results for RØ. Power has been derived from the power curve for an SWT2.3-93 turbine (Fig. 3), the turbine type used for Rødsand II.



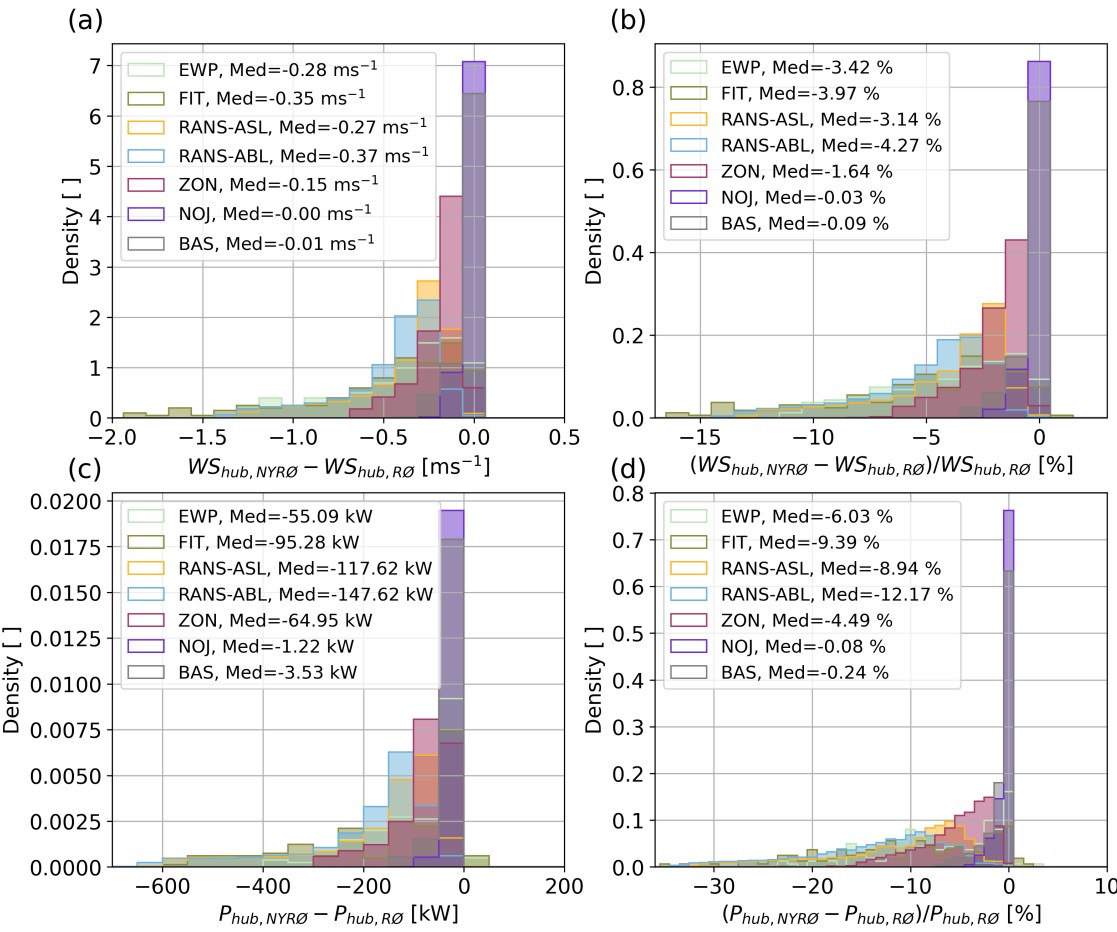

**Figure 14.** Histogram of reduction in (a),(b) wind speed ($WS$) and (c),(d) power ($P$) at hub height in the area of Rødsand II (solid rectangle around Rødsand II in Fig. 2) due to the presence of Nysted in terms of (a),(c) absolute differences and (b),(d) relative differences.

Median reductions in available wind speed resource for Rødsand II due to the presence of Nysted for the flow case of easterly wind around 10 ms$^{-1}$ are about 0.3 ms$^{-1}$ or between 50 kW and 150 kW for EWP, FIT and RANS. This corresponds to a relative reduction in available wind resources of 3 % – 4 % or equivalently to 6 % – 12 % reduction in power output compared to a scenario with a stand-alone Rødsand II wind farm without Nysted wind farm being present.

    As expected from the previous analysis according to the PyWake simulations the influence of Nysted on Rødsand II is much
smaller (median around 0.1 % reduction in wind resources or up to 0.24 % reduction in power output). ZON, being the most complex engineering wake model, shows the largest reductions out of the three chosen wake models. Since the high-fidelity RANS and WRF simulations agreed better with SCADA data Sec. 3.2.2, one can conclude that the chosen PyWake models in the current set-up are not suited to study farm-to-farm wakes.



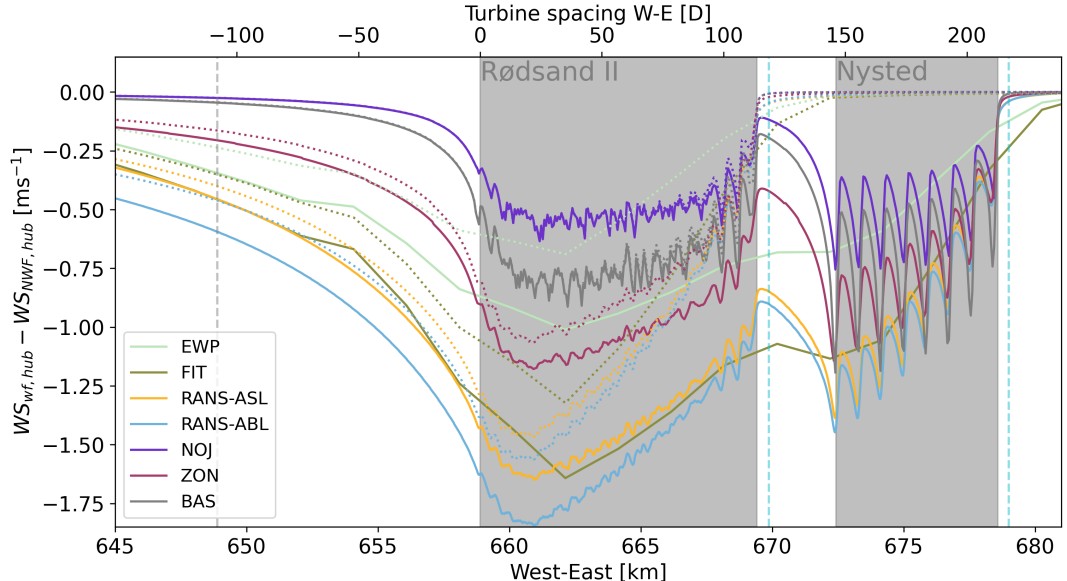

**Figure 15.** West-east cross section of wind speed deficit with respect to a no wind farm (NWF) scenario averaged in north-south direction over the pink area in Fig. 2 for the f1 filter method. NWF corresponds for RANS and PyWake to the eastern most grid points and for EWP and FIT to the NWF simulation. Solid lines correspond to the scenario NYRØ, while dotted lines correspond to the scenario RØ. For abbreviations see Table 7 and Table 5.

Comparing the WRF simulations with the PyWakeEllipSys simulations in more detail, indicates that the median wind speed
reductions due to the presence of Nysted for EWP and FIT lie in between RANS-ASL and RANS-ABL. Thus, concerning estimating median farm-to-farm wake effects both EWP and FIT, provide reasonable results. Since power is more sensitive to wind speed deviations due to the cubed relationship between wind speed and power, for power FIT performs better than EWP when compared to the two RANS simulations.

WRF and RANS reduction distributions have a long tail ranging to up to 15 % reduction available wind resources or
340 equivalently up to 35 % reduction in power output due to the presence of Nysted. In terms of looking at the full distributions, FIT visually agrees better with RANS-ABL than EWP. RANS-ASL results are shifted to smaller reductions compared to RANS-ABL, which is expected due to the pure neutral stratification used in RANS-ASL compared to RANS-ABL, where near-neutral conditions are employed.

To better understand the wake recovery of the different wake models, west-east cross sections with averaged wind speed
deficit in north-south direction over the dashed area in Fig. 2 are shown in Fig. 15 for both NYRØ (solid lines) and RØ (dotted lines) for the different wake models (colours). Wind speed deficit at hub height with respect to a no wind farm (NWF) scenario are shown. As before, the NWF scenario for for RANS and PyWake corresponds to the eastern most grid points, which are undisturbed by the presence of the farms, and for EWP and FIT to the WRF-NWF simulation.





The cross-section highlights again the strong intra-farm variability for RANS-ASL, RANS-ABL, NOJ, ZON and BAS that cannot be captured with FIT and EWP due to the coarser resolution. It shows also that the ZON results compare better with the RANS results for Nysted than the other two chosen wake models of the PyWake suite, since it can capture an accumulated wind speed deficit from east to west in Nysted. However compared to RANS, all PyWake models recover fast and show a much smaller reduction in the Rødsand II area.

Comparing the RANS results with EWP and FIT shows again the better agreement between RANS and FIT for both farms. Overall FIT results lie in between RANS-ABL and RANS-ASL for NYRØ (solid lines) and are therefore considered to well suited to study farm-to-farm effects. EWP shows smaller deficits than RANS in Rødsand II and the results lie in between ZON and BAS. Thus, EWP agrees more with the engineering wake models than with RANS-CFD.

Comparing the results for NYRØ (solid lines) and RØ (dotted lines) indicates again that the effect of Nysted on Rødsand II is minimal for all PyWake engineering models (barely visible at the eastern part of Rødsand II), except for ZON. In contrast, substantial differences in available wind resources of about between 1 ms$^{-1}$ at the inflow to Rødsand II and about 0.25 ms$^{-1}$ towards the outflow of Rødsand II exist for RANS and WRF. Thus, in addition to a distribution of resource reductions due to the presence of Nysted (Fig. 14), the reductions differ spatially with larger reductions close to Nysted. This is expected, since Nysted's influence is strongest close to the farm itself. However, NYRØ and RØ still differ far more west than Rødsand II.

To quantify this difference in long-distance wakes, wind speed deficits with respect to a no wind farm scenario are quantified for NYRØ and RØ 10 km downstream of Rødsand II in Table 8. This distance has been chosen, since it is still relatively undisturbed from the island of Fehmarn, which affects the WRF results further downstream (dashed area in Fig. 2) and is not considered in the PyWake and PyWakeEllipSys simulations. For WRF, a weighted average based on distance of the closest two grid points is used, while for PyWake and PyWakeEllipSys the closest grid point is chosen, which is close enough due to the high resolution of those simulations.

Comparing the wind speed deficit 10 km downwind of Rødsand II (Table 8) indicates that while the NOJ and BAS show almost no wind speed deficit (0.03–0.05 ms$^{-1}$ or about 0.3-0.5 %), ZON, EWP, FIT and RANS still show deficits of 0.2-0.6 ms$^{-1}$ (2-6 %). Comparing the difference between NYRØ and RØ at the same location based on two significant digits, no difference for NOJ and BAS is visible underlining the rapid wake recovery of these two models. ZON still shows at difference due to the presence of Nysted of 0.5 % 10 km downstream, i.e. half of the difference for RANS and WRF (0.1 ms$^{-1}$ or 1 %).

This shows that wind resources of wind farms more that 25 km upstream are still affect by about 1 % for both RANS and WRF simulations. This indicates that farm-to-farm wakes are important to consider for more than 25 km for wind speed ranges just below rated wind speed even for neutral conditions. This agrees relatively well with the flight measurements presented in Cañadillas et al. (2020), although with some uncertainty, since they only show point measurements for three neutral and unstable cases.

### 3.4 Comparison of global blockage effect modelling

The main focus of this study is the comparison of wake modelling for different classes of models. However, it also gives the opportunity to look at the modelling of global blockage effect ahead of Rødsand II and Nysted. To do so, the wind speed deficit



**Table 8.** Wind speed deficit at the closed grid point (for PyWake and PyWakeEllipSys) and interpolated (for WRF) for each model to 10 km east of Rødsand II (dashed gray line in Fig. 15) for the different wake models. All results are averaged in north-south direction over the pink area in Fig. 2.

|          | EWP            | FIT            | RANS-ASL       | RANS-ABL       | NOJ            | ZON            | BAS            |
| -------- | -------------- | -------------- | -------------- | -------------- | -------------- | -------------- | -------------- |
| NYRØ     | -0.35 (-3.52 %) | -0.46 (-4.67 %) | -0.46 (-4.55 %) | -0.60 (-5.93 %) | -0.03 (-0.25 %) | -0.21 (-2.03 %) | -0.05 (-0.45 %) |
| RØ       | -0.24 (-2.42 %) | -0.35 (-3.59 %) | -0.38 (-3.81 %) | -0.45 (-4.47 %) | -0.02 (-0.24 %) | -0.16 (-1.57 %) | -0.04 (-0.43 %) |
| NYRØ-RØ  | -0.11 (-1.10 %) | -0.11 (-1.08 %) | -0.07 (-0.74 %) | -0.15 (-1.46 %) | -0.00 (-0.01 %) | -0.05 (-0.46 %) | -0.00 (-0.02 %) |

**Table 9.** Wind speed deficit 5 rotor diameter ahead of Nysted (for NYRØ) and 5 rotor diameter ahead of Rødsand II (for RØ) for the different wake models. Results are taken from the closed grid points for PyWake and PyWakeEllipSys and are interpolated to 5 rotor diameter for WRF. All results are averaged in north-south direction over the pink dashed area in Fig. 2.

|            | EWP            | FIT            | RANS-ASL       | RANS-ABL       | NOJ            | ZON            | BAS            |
| ---------- | -------------- | -------------- | -------------- | -------------- | -------------- | -------------- | -------------- |
| Nysted     | -0.12 (-1.19 %) | -0.24 (-2.28 %) | -0.04 (-0.39 %) | -0.04 (-0.39 %) | -0.01 (-0.08 %) | -0.02 (-0.21 %) | -0.01 (-0.08 %) |
| Rødsand II | -0.09 (-0.88 %) | -0.19 (-1.81 %) | -0.03 (-0.29 %) | -0.03 (-0.32 %) | -0.01 (-0.07 %) | -0.02 (-0.17 %) | -0.01 (-0.07 %) |

with respect to a no wind farm scenario is investigated at the 5 rotor diameters upstream of each farm. Due to the different turbine types of the two farms (Table 1), this is 412 m upstream for Nysted and 465 m for Rødsand II and is shown as light blue dashed lines in Fig. 15. As for the analysis of the long-distance wake effect, WRF results are calculated from an average of the closest grid points weighted by distance to the point of interest, while for PyWake and PyWakeEllipSys the closest grid points in north-south direction are used. To investigate the global blockage effect of Nysted the NYRØ scenario is used, whereas for Rødsand II the RØ scenario is used, since in NYRØ also the wake effect of Nysted is present. The results for the different models are summarized in Table 9.

The global blockage model selected for PyWake, SSD (Sec. 2.2.2), shows the smallest blockage effect (<1 %), followed by the RANS simulations (between 0.3 and 0.4 %) and WRF simulations (between 0.9 and 2.3 %). Consistent across all models, the blockage affect is smaller for Rødsand II than for Nysted. This could be due to the different shape or being an artifact of the chosen average area. Based on the very few existing blockage measurements, negligible blockage is expected for neutral and unstable conditions as shown in Schneemann et al. (2021) from scanning lidar measurements. In stable conditions the global blockage effect can amount to 2-6 % (Schneemann et al., 2021). The global blockage effect for WRF is likely overestimated, since a wake-affected grid point and a non-wake-affected are averaged here to extract the results at a similar location as for PyWake and PyWakeEllipSys. Due to the coarse resolution of WRF, which assumes the same wind speed for the entire 2 by 2 km area, this result is very much grid dependent and could change for a slightly different WRF grid. Thus, these results should be treated qualitatively and not quantitatively.



## 4 Discussion and conclusion

Simulations for the two neighbouring wind farms Nysted and Rødsand II in the Fehmarn Belt area are performed with models of different complexity, fidelity, scale and computational costs. The results are compared for the simulation of intra-farm and farm-to-farm wakes as well as for the simulation of wake recovery and global blockage effect both against each other and against SCADA data. The model complexity ranges from mesoscale model simulations with two different wind farm parameterizations (WRF-EWP and WRF-FIT), over high-resolution CFD-RANS model simulations equipped with an actuator disk model using PyWakeEllipSys employing two different inflow models (RANS-ASL and RANS-ABL), and three rapid engineering wake models included in the PyWake suite (NOJ, ZON, BAS).

The analysis of the real WRF simulations for a one month period in October 2013 indicated that the WRF model can well represent the meteorological conditions for that period. Comparing to wind turbine wake influenced mast measurements of wind speed and direction, FIT performs better than EWP. However, EWP still improves the evaluation compared to a simulation without wind farms.

Comparing the filtered WRF results to a farm-to-farm flow case that has been simulated with PyWake and PyWakeEllipSys shows that, as expected from the coarse mesoscale resolution, FIT and EWP cannot capture peak wind speed deficits downstream of individual turbines. However, especially FIT compares well with average wind speed reductions simulated by PyWakeEllipSys. EWP underestimates the wind speed deficits compared to the RANS simulations. However, both EWP and FIT perform well for estimating median wind resource reductions due to the presence of a upstream farm. Expected median reductions in power output are better represented in FIT than in EWP when compared to RANS.

The engineering wake models in the PyWake suite agree reasonable well with the RANS simulations within Nysted and at the its outflow. However, all engineering models simulate a much more rapid wind deficit recovery downstream of the wind farms and thus a one order of magnitude smaller farm-to-farm wake effect. Considering the agreement of WRF and PyWakeEllipSys and the their agreement with the SCADA data, all considered engineering wake models likely underestimate the wind farm wake effect. Thus, the results of these wake models should not be trusted when farm-to-farm wakes are an issue, while they can well represent intra-farm wakes. However, using a more sophisticated deficit model like ZON, can improve the prediction of a farm-to-farm effect compared to simpler models.

The analysis of the long-distance wakes showed that even in neutral conditions, the effect of an upstream farm can still amount to a wind resource reduction of 1 % 25 km downstream. This is consistent for both RANS and WRF simulations and indicates that farm-to-farm effects should not be neglected for wind resource planning offshore.

The study also shows challenges, when comparing idealized flow cases with real data: The NWF simulations indicated that wind direction over the study area is likely influenced by coastal effects. This leads to spatial variability in the flow, especially with respect to wind direction over this 40 km area of interest. More measurements are required to verify this wind direction variability across the domain of the two wind farms. However, this still indicates that for offshore farms close to the coast uniform inflow flow cases cannot capture the variability due to local effects downstream. Thus, for more accurate high-resolution wind simulations, these effect should be accounted for when applying high resolution models in the future.





The flow case selected for this study represents a situation with strong influence of wakes, since a wind speed range just
below rated wind speed has been chosen. The results could be extended by considering other wind speed ranges to generate
a more complete picture. The selection of the simulation period was constraint by the available measurements. However, due
to the infrequent occurrence of easterly flow, only very few relevant time stamps could be identified. Thus, the results contain
some uncertainty. Other periods as well as the evaluation of wind farm pairs with other layouts should be investigated in further
studies to confirm the results from this study.

In this study, we applied a TKE factor of 0.25 in the FIT parameterization (Table 4) following the recommendations by
Archer et al. (2020). Based on our simulations, this seems to be a reasonable choice as the FIT simulations agree well with the
RANS simulations. However, since we only tested this coefficient and focused only on the evaluation of wind speed deficit, we
cannot conclude that 0.25 is the best choice in all conditions. Larsén and Fischereit (2021) for instance compared WRF-FIT
simulations against flight measurements and found that for their simulation a TKE factor of 1 was more appropriate. Thus,
more studies are needed to derive the best TKE factor.

This studies indicated that average intra-farm wakes can be captured reasonably well with a resolution of 2 km, a typical
resolution of mesoscale models for wind energy applications according to the review by Fischereit et al. (2021a). A higher
resolution of e.g. 1 km could be tested to see whether the agreement improves even more.

Considering the three main aims of this study, we can conclude that average intra-farm variability can be captured reasonable
well with the Fitch et al. (2012) wind farm parameterization using a resolution of 2 km, while EWP underestimates wind speed
deficits. The considered RANS model, PyWakeEllipSys, can simulate wind farm wakes and long distance wakes well, but even
for relatively small areas offshore uniform inflow conditions are not always met near the coast. Thus, if long distance wakes
are of interest, effects of terrain and roughness should be considered in future PyWakeEllipSys simulations for coastal areas
to provide a more realistic picture. However, mesoscale phenomena, like sea breezes, cannot be accounted for in CFD-RANS
models, which is the strength of using mesoscale models like WRF. All considered engineering wake models from the PyWake
suite underestimate the wind farm wake effect, although with different magnitudes. The most complex out of the here applied
engineering deficit models, ZON, provides the best option to account for a farm-to-farm effect out of the tested rapid models.
However, it still underestimates the median wind resource reduction due to an upwind farm by about 50 %. Therefore, overall,
the higher computational costs of PyWakeEllipSys and WRF pay off in terms of accuracy for farm-to-farm wake modelling
compared to rapid models like PyWake.

*Code and data availability.* The WRF source code is available from https://github.com/wrf-model/WRF. ERA5 input data for WRF is available from Hersbach et al. (2018). The OSTIA data is available from http://my.cmems-du.eu/motu-web/Motu. The EWP source code is available upon request and will be made available in the future under https://gitlab.windenergy.dtu.dk/WRF/EWP. PyWake is available from https://gitlab.windenergy.dtu.dk/TOPFARM/PyWake. The configuration files for WRF and PyWake are available from Fischereit et al. (2021b). PyWakeEllipSys documentation is available from https://topfarm.pages.windenergy.dtu.dk/cuttingedge/pywake/pywake_ellipsys/index.html.



*Author contributions.* JF, KH, XL and PL designed the experiments. JF conducted the WRF experiments and JF and XL post-processed the results. PL conducted the PyWakeEllipSys simulations and post-processed the results. JF conducted the PyWake simulation with help of PR and JL. KH processed the measurements and conducted the WRF filtering. JF analysed and visualized the simulation results. JF prepared the manuscript with contributions from all co-authors.

*Competing interests.* The authors declare that they have no conflict of interest.

*Acknowledgements.* This study was partly funded by ForskEL/EUDP through the OffshoreWake project (PSO-12521/EUDP 64017-0017), by Independent Research Fund Denmark through the 'Multi-scale Atmospheric Modeling Above the Seas' (MAMAS) project (nr. 0217-00055B), by the DTU TOPFARM project and by the project 'ModFarm: Advanced multi-objective design tool for wind farms'. Data processing and visualization for this study was in part conducted using the python programming language and involved use of the following
software packages: NumPy (van der Walt et al., 2011), pandas (McKinney, 2010), xarray (Hoyer and Hamman, 2017), Matplotlib (Hunter, 2007). The colors for the line plot have bee selected through the "Color Cycle Picker" at https://github.com/mpetroff/color-cycle-picker (last access: 29 May 2021). The authors are grateful for the tools provided by the open-source community.





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
