# Peer review of "Comparing and validating intra-farm and farm-to-farm wakes across different mesoscale and high-resolution wake models"

_Wind Energy Science, 2021_

## Author Comment (AC1)

**Response to the comments about the submitted paper**

**Comparing and validating intra-farm and farm-to-farm wakes across different mesoscale and high-resolution wake models**

We would like to thank the reviewers for the useful comments and suggestions. Our detailed answers follow.

Please note that reviewers' comments are in italics while our answers are not. Additions to the original manuscript are indicated in blue.

**Answers to Reviewer 1**

**Comment R1.1** *Throughout the manuscript, please update legends where necessary (e.g. Fig 3). WES guidelines state "A legend should clarify all symbols used and should appear in the figure itself, rather than verbal explanations in the captions (e.g. "dashed line" or "open green circles")." https://www.wind-energy-science.net/submission.html*
**Answer to R1.1** Thank you for pointing this out. We have missed these instructions. We have now reworked several figures to follow the guideline.

**Comment R1.2** *L11 and L449: possibly "reasonably" instead of "reasonable"*
**Answer to R1.2** Thanks for pointing this out. We adjusted it in L11 and L449 as well as in L418.

**Comment R1.3** *L11-13: Please quantitatively state the bias of both Fitch and the EWP. The Fitch wind speed bias is only 0.12 m/s (or ~15%) smaller than the EWP bias, but based on the current writing, I assumed they had a much larger difference in my first readthrough*
**Answer to R1.3** In the abstract, we referred to the comparison against RANS simulations (i.e. figure 11 and 12 of the old manuscript), where EWP clearly underestimates the deficit. To make that clear, we extended the abstract as follows "Based on the performed simulations, we can conclude that both WRF+FIT (BIAS = 0.52 ms$^{-1}$) and WRF+EWP (BIAS=0.73 ms$^{-1}$) compare well with wind farm affected mast measurements. Compared to the RANS simulations, average intra-farm variability can be captured reasonably well with WRF+FIT using a resolution of 2 km, a typical resolution of mesoscale models for wind energy applications, while WRF+EWP underestimates wind speed deficits.". Please note that these numbers are slightly changed compared to the old manuscript. This is, because we decided with respect to your comment 15 that it would be better to interpolate WRF results to the mast grid point instead of taking the closest grid point. That is what we have done for the new numbers.

**Comment R1.4** *L14-15: Please clarify that you are refering to peak wind speed deficits (because average deficits differ between PyWake and RANS)*
**Answer to R1.4** Indeed, thanks for spotting this missing information. We have added the term "peak". The new version now reads "All considered engineering wake models from the PyWake suite simulate peak intra-farm wakes comparable to the high fidelity RANS simulations."

**Comment R1.5** *L36: "Pay off" is somewhat ambiguous*
**Answer to R1.5** We modified it as follows: "Different model types do not only differ in the typical domain size and therefore in the spatial scales that they can capture, but also by their computational costs (Fig. 1). Thus, based on the analysis this study also addresses the question, whether more computational demanding model simulations provide better accuracy when modelling intra-farm and farm-to-farm wakes.".

**Comment R1.6** *L40-41: Fishereit et al. (2021a) detail a vast number of WRF WFP studies that simulate wakes inside of a farm (e.g. Shepherd et al. (2020) compares FIT and EWP). I am confused by the statement "only one study... evaluated intra-farm wakes with EWP". Perhaps a different definition from "intra-farm" is being used than as I understand it.*
**Answer to R1.6** Indeed, that was not written very clearly. We meant the evaluation of intra-farm wakes against observations. We have modified the sentence as follows "only few studies (Hansen et al., 2015; Poulsen, 2019) evaluated intra-farm wakes with EWP against measurements

or SCADA data".

**Comment R1.7** *L54-55: What is a "high-resolution" wake model?*
**Answer to R1.7** Thanks for pointing out that this is not a commonly used term. To make it clear we added in L33 in the new manuscript, where the term is used for the first time, some explanation: ... and high-resolution wake models, sometimes also termed wind turbine wake models (Göçmen et al., 2016),...

**Comment R1.8** *L70: While I understand why the power and trust curves are important, why is the rotor speed curve important?*
**Answer to R1.8** The rotor speed curved is used in PyWakeEllipSys to model to thrust and tangential forces on the actuator discs. The shape of the force distribution is a function of the tip speed ratio and the magnitude of the tangential force is dependent on the rotor speed. This has been added to the manuscript as follows in Section 2: Note that the rotor speed is only used for the PyWakeEllipSys RANS-AD simulations where it influences the shape of the thrust and tangential force distributions; and it defines the magnitude of the tangential force that induces wake rotation, see 2.2.1 for more details and (which induces a small wake deflection when combined with a wind shear) and in Section 2.2.1: The shape of the force distributions is a function of the thrust coefficient and tip speed ratio, and the magnitude of the tangential force depends on the rotor speed and power coefficient.

**Comment R1.9** *L104,L106: Missing parenthesis*
**Answer to R1.9** Thanks for spotting! We added it accordingly.

**Comment R1.10** *L117: Is there a particular reason to aim for 7% TI at hub height?*
**Answer to R1.10** The value has been chosen based on a on-site measurements from the metmast in combination with a previous study. We clarified this by adding: "The TI value of 7% was chosen based on a directional analysis of measured streamwise TI, from cup anemometers in 48 m height at the RØ mast for 11.2002 — 06.2005, i.e. before the installation of Rødsand II (Fig. 6). Differences in TI between sectors are visible, reflecting landward and seaward sectors (Fig. 2). For easterly wind, a streamwise TI of 10.4% is estimated. This value is likely influenced by the presence of Nysted. The closest seaward sector, 120°, has a streamwise TI of about 7%. Converting both streamwise TI values, $TI_u$, to total TI using the approach by Panofsky and Dutton (1984), $TI = \sqrt{1/3(1 + 0.8^2 + 0.5^2)}TI_u$, one arrives at 8% and 6 %, respectively. These values are very similar to the TI value of 7%, chosen in Hansen et al. (2015) for a wind speeds around 8 ms$^{-1}$. Thus, 7% has been chosen here for simplicity. A slight sensitivity of the results to the chosen TI value can be expected.". The following figure is added to the manuscript:

[Figure]

**Comment R1.11** *L121: What is meant by "in full balance"? Time-varying WRF models show blockage effects, so steady-state is not required to study this effect*

**Answer to R1.11** Both steady and unsteady models can simulate blockage effects (or wind farm induction); however, it can be challenging to estimate the wind farm induction if the freestream wind speed is varying, especially for small quantities as wind farm induction. Hence, a varying freestream wind speed may influence the results; this also applies to field measurements. A common practical approach for unsteady numerical models is to perform two simulations, one with and one without wind turbines forces. However, this can still lead to errors because an error in the intended inflow wind speed could also lead to different thrust coefficients.

RANS is a steady-state model and this makes it more easy to use for blockage studies. However, if the inflow is not in full balance with an empty domain, then the inflow will develop downstream and this will affects the freestream wind speed that the wind turbines will experience. In that case, the RANS model becomes dependent on the distance between the inlet and wind farm, which is not desired. To avoid this, we perform a 1D precursor simulation of all inflow models in order to assure that the inflow is in full balance with an empty 3D domain.

In addition, we are not sure if WRF can properly model wind farm induction due to the large horizontal cell size, as we state in line 395ff of the original manuscript.

**Comment R1.12** *L191: What is a Gaussian average?*

**Answer to R1.12** Wind direction uncertainty is the main factor that needs to be considered when comparing wake model simulations with measurements (Gaumond et al., 2014). A Gaussian average (or Gaussian convolution) is applied in order to predict the mean power in a wind flow case given a Gaussian distributed wind direction 10-minute fluctuations. We have added the reference to the manuscript As a post processing step, the flow variables are Gaussian averaged (Gaumond et al., 2014) over the wind directions with a standard deviation of 5°....

**Comment R1.13** *Sec 3.1: This doesn't necessarily have to be mentioned in the manuscript, but I believe a month-long WRF WFP comparison to met mast data is the longest WFP-to-mast comparison to date*

**Answer to R1.13** Thanks for acknowledging the WFP-to-mast comparison performed in this study.

**Comment R1.14** *Figure 8: I am having a tough time understanding the vertical red and yellow bars. Do these mark the 9 and 17 timesteps with data?*

**Answer to R1.14** We are sorry about the confusion. We have clarified the description as well as modified the plot, which now uses arrows instead of vertical lines and shows $t_{f1}$ and $t_{f2}$ instead of f1 and f2. The description now reads "The vertical arrows mark the time steps $t$ of the two filter methods f1 (red) and f2 (yellow) ..."

**Comment R1.15** *L237: In WRF WFP validation studies, we always struggle with two interrelated questions. (1) How accurately does WRF simulate the background flow? (2) How accurately does WRF model wake effects? Because the observational mast is very close to a turbine, it is technically not possible to truly calculate WRF's bias in background flow. However, it is possible to characterize bias during "strong waking" (as is done in L247) and "weak waking". Please quantify bias under these two conditions, defined as you see fit.*

**Answer to R1.15** We agree with the reviewer on that, through the comparison with mast measurements, it is difficult to quantify WRF's ability in capturing the background flow, as we have only one "realization" in the measurements, which is clearly influenced by the presence of the farm in our case. However, in the simulations, such a bias is the same in both NWP and WRF-WFP/EWP, which makes it possible to assess the wind farm impact. The comparison between measurements and the simulation with/without wind farm effects can therefore give us an overall evaluation of the quality of wake calculation. However, we also follow your suggestion and quantified the wake effect under strong and weak waking as follows: Performing the same analysis filtered for eastern wind directions (90°±10°) at the RØ mast, but without filtering for wind speed, indicates a considerable larger wind farm effect based on 127 10-minute values: $WS_{\mathrm{bias,NWF}}$ =1.04 ms$^{-1}$, $WS_{\mathrm{bias,EWP}}$ =0.32 ms$^{-1}$ and $WS_{\mathrm{bias,FIT}}$ =-0.14 ms$^{-1}$, which amounts to a wind farm effect of 0.73 ms$^{-1}$ for EWP and 1.2 ms$^{-1}$ for FIT at 57 m height that can be largely attributed to the wake effect of the farms. Selecting non-wake-affected sector between 260° and 30° for comparison, shows that the bias is strongly reduced for the NWF scenario: $WS_{\mathrm{bias,NWF}}$ =-0.02 ms$^{-1}$, $WS_{\mathrm{bias,EWP}}$ =-0.17 ms$^{-1}$ and $WS_{\mathrm{bias,FIT}}$ =-0.28 ms$^{-1}$, which would correspond to a wind farm effect of -0.15 ms$^{-1}$ for EWP and -0.25 ms$^{-1}$ for FIT. The larger biases for EWP and FIT compared to NWF for the non-wake-affected sector can be partly attributed to the interpolation of wake-affected and non-wake-affected points to the mast location.

**Comment R1.16** *L242: Please quantify "medium high wind speeds"*

**Answer to R1.16** We agree that this is not very precise. We modified the text, which now reads "and FIT show the largest improvements around 8–13 ms$^{-1}$"

**Comment R1.17** *Figure 10: The small map showing the location of the transect is an excellent feature*

**Answer to R1.17** Thanks.

**Comment R1.18** *L267: In contrast to what?*

**Answer to R1.18** We modified the sentence, which now reads "While the inflow direction agrees better between models for f1, f2-filtered SCADA data agree better with RANS-ABL compared to f1."

**Comment R1.19** *Sec 3.2.2: "Average deficits" are often discussed, but they are not quantified anywhere. Please share these values throughout this subsection.*

**Answer to R1.19** That was probably a bad choice of words from our side. We were referring to the wind speed deficit in between turbines, in contrast to the peak deficit at the turbines. We now changed the term and also added an explanation to the text However, especially WRF-FIT compares very well with the baseline deficit of RANS-ASL and RANS-ABL, i.e. the wind speed deficit in between of turbines, for all transects for f1. We changed the term throughout the manuscript.

**Comment R1.20** *Figure 11: It is difficult to distinguish between the models within each of the spikes. For example, I believe there should be orange spikes in panel b, but I do not see any. Consider shifting some of the models downward on the y-axis by 50D and note that this shift is being done to help improve legibility*

**Answer to R1.20** Thanks for the suggestion. Rather than shifting some models downward, we decided to use subplots. In this way, there is no confusion about the wake position in south-north direction. We also added error-bars around the SCADA data to show the variability. The updated figure looks like this

[Figure]

**Comment R1.21** *L304: Please roughly quantify "close to turbines"*
**Answer to R1.21** We replaced this by peak wind speed deficits.

**Comment R1.22** *Figure 14: Thank you for showing both absolute and relative differences*
**Answer to R1.22** Thanks.

**Comment R1.23** *L329: Consider reminding the reader that these reductions are focused on only 10 m/s winds for this inflow direction*
**Answer to R1.23** We added "for a flow case of around 10 ms$^{-1}$"

**Comment R1.24** *Figure 15: Please add an arrow that points to the left to remind readers that the wind goes in that direction*

**Answer to R1.24** We added such an arrow to the figure:

[Figure]

**Comment R1.25** *L410: Please quantify the relative performance of FIT and the EWP*
**Answer to R1.25** We have modified to the sentence, which now reads "Comparing to wind farm wake influenced mast measurements of wind speed, FIT (WS-bias: 0.52 ms$^{-1}$) performs better than EWP (WS-bias: 0.73 ms$^{-1}$). However, EWP still improves the evaluation compared to a simulation without wind farms."

**Comment R1.26** *L416: Please clarify "perform well relative to ?"*
**Answer to R1.26** We meant that both WFPs are well able to estimate the median wind resource reduction. We modified the text as follows to make it more clear "However, both EWP and FIT can well estimate median wind resource reductions due to the presence of a upstream farm."

**Comment R1.27** *L418-424: This paragraph reads very well*
**Answer to R1.27** Thanks.

**Comment R1.28** *L438: Thank you for urging some caution due to the small amount of data*
**Answer to R1.28** Thanks for mentioning it.

**Comment R1.29** *L449: It was nice to see that the three main aims from the intro were returned to. I will say, the 3rd aim (computational cost) feels like it is not discussed much throughout the paper. I would consider stating that this paper has two main aims.*
**Answer to R1.29** We agree that computational costs are not much discussed within the paper, but serves mainly as an additional motivation. We therefore decided to follow the reviewer's suggestion and replaced the "three aims" by "two aims".

**Comment R1.30** *L475: Thank you for citing the Python packages!*
**Answer to R1.30** Thanks for mentioning it.

**Answers to Reviewer 2**

**Comment R2.1** *Overall comment: This work compares and validates the intra-farm and farm-to-farm wakes across different mesoscale and wake models. I think the topic is certainly interesting and relevant to the wind energy community. In particular, I enjoy reading the part where the authors applied a filtering method to compare WRF with the engineering model. However, more details about that approach are needed. Overall, I think the paper can be accepted for publication after addressing the following comments.*
**Answer to R2.1** Thanks.

**Comment R2.2** *Line 8: It is weird to abbreviate Wind Farm Parametrization to FIT. Maybe rewrite to as Fitch scheme. That would make more sense.*
**Answer to R2.2** Thanks for this suggestion. We chose FIT to distinguish it from EWP, the other wind farm parameterization in our paper. Using WFP to abbreviate the wind farm parameterization developed by Fitch et al. (2012), may confuse the readers, since two wind farm parameterizations are actually applied.

**Comment R2.3** *Lines 199-200; Lines 221-222, can the authors provide more detailed (or an example) about the "averaged weighted according to the WRF-NWF wind speed at the WRF inflow grid point" using information from Table 6? I am not sure what it really means. In addition, I thought only the WRF and SCADA data are subjected to filtering, does the simulations from wake models also undergo some kind of filtering as well based on the quoted sentence.*
**Answer to R2.3** We apologize for the confusion. We have revised the respective paragraph, which now reads: The described filtering for stability, wind speed and wind direction is applied to both SCADA data and the WRF results to identify $N$ 10-minute periods (Table 6), which are comparable to the simulated flow cases for PyWake and PyWakeEllipSys. Using the inflow wind speed for WRF-NWF for each 10-minute period, the two closest simulated PyWake and PyWakeEllipSys inflow cases, i.e. rounded up (called $WS_{fl,u}$ below) and rounded down (called $WS_{fl,l}$ below), respectively, are identified and a weighted average $\overline{WS}_{fl,ave}$ of the two corresponding simulations is calculated. The weight is calculated from $w = 1 - (WS_{69,in,i} - WS_{fl,l})$, where $WS_{69,in,i}$ is the WRF-NWF wind speed at the WRF inflow grid point (Fig. 2) for each period $i$. Using these weights, the average weighted flow case wind speed corresponding to the 10-minute periods is then derived from

$$\overline{WS}_{fl,ave} = \frac{\sum_{i=1}^{N}(WS_{fl,l} \cdot w + WS_{fl,u} \cdot (1-w))}{N}. \tag{1}$$

The WRF results and SCADA data for the identified 10-minute periods are linearly averaged. The corresponding inflow wind speed for the two filter methods are listed in Table 6.

**Comment R2.4** *Wind direction filtered (Table 6; lines 214-215): Since Rodsand II mast is largely influenced by the wake, I am not sure how valid is f2 for wind direction filter. Is there any coincide periods between f1 and f2?*
**Answer to R2.4** Thanks for the comment. We agree that f2 is influenced by the wake, but due to lack of other measurements, this was the only option for us in this study. The two filter methods have one 10-minute period in common. 11 other 10-minute periods in f2 are within 4 hours after 3 identified periods in f1. Considering this and acknowledging the lack of other undisturbed measurements, we still think f2 is a useful addition to the analysis.

**Comment R2.5**  *Lines 237-242: I think it stresses the value of the added TKE in the WFP. Authors could consider adding that in their discussion.*
*Reference:*

*Xia, G., Zhou, L., Minder, J.R. et al. Simulating impacts of real-world wind farms on land surface temperature using the WRF model: physical mechanisms. Clim Dyn 53, 1723–1739 (2019). https://doi.org/10.1007/s00382-019-04725-0*
*Tomaszewski, J. M. and Lundquist, J. K.: Simulated wind farm wake sensitivity to configuration choices in the Weather Research and Forecasting model version 3.8.1, Geosci. Model Dev., 13, 2645–2662, https://doi.org/10.5194/gmd-13-2645-2020, 2020.*

**Answer to R2.5** Thank you for theses suggestions. We are aware of these valuable publications. However, in this context, we feel that we cannot draw the conclusion that it is the added TKE that improves the performance of the wind farm parameterization. EWP and FIT differ not only with respect to an added TKE source, but also due to subgrid-scale wake expansion that is considered in EWP but not in FIT. The added TKE source in FIT likely plays a role here, but more investigations are required to draw a clear conclusion. However, adding or neglecting a TKE source in the two WFPs is beyond the scope of this study.

**Comment R2.6** *Figure 9: There should be more spacing between the top and bottom panels to avoid confusion.*
**Answer to R2.6** We agree with this issue. The spacing has been increased.

**Comment R2.7** *Line 256: what about the results from RANS-ASL?*
**Answer to R2.7** We assume that you refer to line 259 or Figure 10 with this comment. To improve the visibility in this figure only selected simulations are shown. The results for other simulations can be seen in Figure 11 and Figure 12. This has been added to the text as follows:
For readability, only the simulation results for WRF-NWF, WRF-FIT and RANS-ABL are shown; the simulation results for other simulations are shown in Fig. 11 and Fig. 12.

**Comment R2.8** *Missing comma (eg., Line 299: In addition, the deficits are also narrower in south-north direction)*
**Answer to R2.8** Thanks for spotting this. The comma has been added.

**Answers to Reviewer 3**

**Comment R3.1** *The paper uses SCADA data from an offshore wind farm as well as measurements from a co-located met mast to validate different wake models in the situation where the wind farm is affected by wakes from an upstream neighbour wind farm. Both the intra-farm and farm-to-farm wakes are modelled using three different types of models. (1) A mesoscale model that includes details of the regional flow with two different parameterizations of the wind farms. (2) A RANS model that describes the turbine interactions in greater detail that the mesoscale model, but in contrast relies on more idealised inflow conditions. (3) Three engineering wake models. The models are compared to data and to each other. The authors conclude that one of the mesoscale wind farm parameterizations (Fitch) and the RANS simulations agree reasonably well with the data, while the other mesoscale parameterization (EWP) and the three engineering models underestimate the farm-to-farm wake loss.*

*The topic is of great interest considering the plans for an extensive buildout of offshore wind capacity around the world. The paper is easy to follow, and the conclusions are sound given the available data. The analysis does only cover a limited period of one month. I assume that limited computational resources for the mesoscale simulations were the driving factor for this limitation, but the authors should make this clear, since the analysis would be much stronger if a longer period had been analysed. The criteria for selecting the specific month were high data availability for both met mast and SCADA data. Unfortunately, this specific month saw only a limited number of flow cases consistent with the farm-to-farm interactions that form the core of the study. In my view, the met mast data are not as interesting as the SCADA data, so it is possible that a better month could have been chosen based on high SCADA data availability combined with a higher frequency of useable flow cases. It may even have been better to not run the models on a specific calendar month but rather for specific days or weeks to maximize the number of useable flow cases. Nonetheless, the paper is still a solid and important contribution and I enjoyed reading it.*

**Answer to R3.1** Thank you for your evaluation. We agree with the reviewer that larger data base would form better statistics. We hope in the near future we could extend the current study to longer study period, where we shall also address how long is long enough, in the context of the representativeness of the dataset. Such an action is however limited at this moment due to our access to the SCADA data and resource. Nevertheless, we also agree with the reviewer that this limited amount of analysis can, on its own, already provide valuable information on the performances of the various models. We would like to share these findings with the readers, and at the same time, when drawing the conclusions, we address the uncertainty in relation to limited amount of data.

**Comment R3.2** *Figure 1- consider making this figure larger for better readability*
**Answer to R3.2** We increased the figure size to the text-width.

**Comment R3.3** *Line 41 – consider including a reference to* `https: // zenodo. org/ record/ 3637944/ files/ 1. 7_ Poulsen%20Validation%20of%20wind%20farm%20parametrisation%20in% 20WRF%20using%20wind%20farm%20data%20%28Thesis%29. pdf`
**Answer to R3.3** Thank you for the suggestion. We have added it in the following way "only few studies Hansen et al. (2015); Poulsen (2019) evaluated intra-farm wakes with EWP against measurements or SCADA data".

**Comment R3.4** *Line 43: split "with a"*

**Answer to R3.4** Thanks for spotting! We changed that accordingly.

**Comment R3.5** *Line 75: I don't understand the remark about the effect of wake rotation on the velocity deficit. Consider if it is necessary or if additional explanation can be added.*

**Answer to R3.5** The rotor speed curve of the Bonus wind turbine is uncertain, but this quantity mainly affects the magnitude of the tangential force that induces wake rotation. Wake rotation and shear can deflect the wake slightly, but this effect is small in neutral conditions, where the wind veer is expected to be small (the presence of a large wind veer can strongly influence the effect of wake rotation on wake deflection). In van der Laan et al. (2015a) [Fig. 15], numerical simulations of a row of wind turbines, with and without tangential forces, indicate that the effect of wake rotation on the wake deficit is very small for neutral conditions. This has been added to the manuscript as follows in Section 2: Note that the rotor speed is only used for the PyWakeEllipSys RANS-AD simulations where it influences the shape of the thrust and tangential force distributions; and it defines the magnitude of the tangential force that induces wake rotation, see 2.2.1 for more details and (which induces a small wake deflection when combined with a wind shear) and in Section 2.2.1: The shape of the force distributions is a function of the thrust coefficient and tip speed ratio, and the magnitude of the tangential force depends on the rotor speed and power coefficient.

**Comment R3.6** *Line 117 – Please discuss why a hub height TI of 7% was chosen. Is this based on on-site measurements from the met mast. Does the TI depend a lot on wind direction for this site given the proximity to land in some directions?*

**Answer to R3.6** The value has been chosen based on a on-site measurements from the met-mast in combination with a previous study. We clarified this by adding: "The TI value of 7% was chosen based on a directional analysis of measured streamwise TI, from cup anemometers in 48 m height at the RØ mast for 11.2002 — 06.2005, i.e. before the installation of Rødsand II (Fig. 6). Differences in TI between sectors are visible, reflecting landward and seaward sectors (Fig. 2). For easterly wind a streamwise TI of 10.4% is estimated. This value is likely influenced by the presence of Nysted. The closest seaward sector 120° has a streamwise TI of about 7%. Converting both streamwise TI values, $TI_u$, to total TI using the approach by Panofsky and Dutton (1984), $TI = \sqrt{1/3(1 + 0.8^2 + 0.5^2)}TI_u$, one arrives at 8% and 6 %, respectively. These values are very similar to the TI value of 7%, chosen in Hansen et al. (2015) for a wind speeds around 8 ms$^{-1}$. Thus, 7% has been chosen here for simplicity. A slight sensitivity of the results to the chosen TI value can be expected.". The following figure is added to the manuscript:

[Figure]

**Comment R3.7** *Line 145 – why was the wake decay constant in the Jensen model chosen as k=0.1? This is a very large value given the typical recommended range of 0.03-0.05 for offshore wind farms. Please add a discussion in the text*

**Answer to R3.7** Thank you for reading the manuscript so carefully. Indeed, we made a mistake by not using a typical recommended offshore value. We have rerun NOJ with k=0.04 now and updated the results throughout the manuscript. The overall conclusion does not change that PyWake underestimates wind farm wakes. However, the results for NOJ improved and are now comparable to ZON for Nysted, while they slightly perform worse for Rødsand II and beyond. We modified the text for description as follows The NOJ model has been developed for the far wake and represents the wake as a simple top-hat wake. We apply a linear wake expansion constant of k=0.04, which corresponds to offshore conditions and has been used in previous studies (Nygaard and Hansen, 2016). We also updated the provided PyWake script in our zenodo-repository https://doi.org/10.5281/zenodo.5570396.

**Comment R3.8** *Line 149 – the BAS wake expansion constant is given as k=0.0324555. This may be following the original paper, but it seems like more digits than can be justified. How many decimal places can be seen in the results?*

**Answer to R3.8** The wake expansion comes indeed from the original calibration. We have reduced the number of digits in the text to k=0.03.

**Comment R3.9** *Line 158 – please comment on the grids used in the PyWake engineering model calculations. Are they only used for visualization purposes, or are they an integral part of the calculations of wake effects?*

**Answer to R3.9** PyWake (and other) engineering wake models do not rely on a grid for calculation, as they only predict the flow at each turbine location. You are right that there is grid evaluation used for visualization.

**Comment R3.10** *Page 10 – please indicate the order of magnitude of computation time for all the simulations. This is relevant for a reader interested in performing similar calculations.*

**Answer to R3.10** We added the numbers at the respective places in the manuscript. For WRF Each 5.5 day period took roughly 6 hours of simulation time on 64 cpus.; for PyWakeEllipSys The PyWakeEllipSys simulations took about 1 to 2 hours using 985 cpus for each wind direction, wind

speed, inflow model and wind farm case. The relative high computational is caused by the strict convergence requirement in order to resolve wind farm induction and the large number of cells (258 million) that is partly used to capture the wind farm wakes.; and for PyWake Depending on the chosen wake model and wind farm configuration, each PyWake simulation, i.e. for all wind speed and directions, takes about 1–2 hours to complete on 32 cpus.

**Comment R3.11** *Line 191 – why is the width of the Gaussian wind direction average 5 degrees? Where does this number come from? A reference to Gaumond et al, Wind Energy would be good to add as well.*

**Answer to R3.11** This is a good point. We have added a discussion to Sect. 2.3.1: The chosen value of the standard deviation for the Gaussian averaging is based on the work of Gaumond et al. (2014), where a smaller standard deviation (2.7°) was estimated from met mast measurements, but a larger standard deviation (4.5-7.4°) was necessary in order to fit the modeled power of the individual wind turbine rows of the Horns Rev I wind farm with the measured power that includes wind direction uncertainty. The later is further explained in Section 3.1.3 of van der Laan et al. (2015), where it is shown that the standard deviation increases linearly with the distance from the location at which the inflow wind direction has been measured. Overall, the applied 5° standard deviation in the present work is a rough estimate based on these previous works.

**Comment R3.12** *Figure 7 – The Rødsand II wind farm is abbreviated RS2 in this figure, while in table 5 it is referred to as RØ. It would be better to use a consistent abbreviation.*

**Answer to R3.12** Thank you for spotting this. We changed RS2 in Figure 7 to RØ. We would also like to report that we accidentally swapped the grid point to the east and to the west. Since the point of this plot was to show the difference between the location east and west of the mast, this mistake does not influence the results. This is the updated version of the plot:

[Figure]

**Comment R3.13** *Figure 8 – I can only really make out the black, orange and blue lines (sonic, NWT_in and FIT_e). Consider reducing the number of lines. I cannot know which of the other lines the curves I cannot see are hiding behind.*

**Answer to R3.13** We follow your suggestion and reduced the number of lines to sonic, FIT and EWP that you mentioned. In addition, we decided to interpolate the WRF results to the mast grid point instead of using the closest grid point to the east to have a more fair comparison. The updated figure is as follows

[Figure]

**Comment R3.14** *Figure 8 – The text states that the WRF simulations generally agree with the sonic measurements, with one exception being the storm on 28 October. I would say that there are a few examples where the deviation between WRF and measurements is even larger. I think in general the agreement is good, but you should maybe rephrase this so it doesn't sound like the storm is the only instance of disagreement between simulations and measurements.*

**Answer to R3.14** We agree that there are other instances with deviations. We therefore removed the sentence about the storm. The sentence now reads: "In general, the time series for both $WS$ and $WD$ for $FIT$ and $EWP$ agree well with the sonic measurements with a few exceptions."

**Comment R3.15** *Line 239 – Referring to Figure 9 the text states that Fitch clearly performs best for simulating the wind speed deficit. But Figure 9 does not show the wind speed deficit, only the wind speed. In addition, only a few points in Figure 9 are affected by wakes from Nysted. To support the statement, it would be more relevant with a scatter plot of wind speeds showing only the directions where Rødsand II is in the wake from Nysted.*

**Answer to R3.15** Indeed, the figure does not show the wind speed deficit, but rather the wind speed, where we can, however, see the effect of the wind farm parameterization as a reduced wind speed compared to the simulation without wind farms. We corrected that in the text. Thus, FIT clearly performs best for simulating $WS$ at the mast, the improvements for $WD$ compared to NWF are similar for EWP and FIT. However, we decided to stick with the scatterplot for the full length of October, since it shows a wind farm effect even if not many waked wind direction time stamps are included. However, we follow your suggestion in that regard that we provide biases for the waked conditions. See the response to the next comment for that.

**Comment R3.16** *Line 243 – The estimation of the wind farm wake effect for October 2013 by subtracting the NWF wind speed bias from the FIT or EWP wind speed bias neglects the global blockage effect on the met mast. The mast is quite close to the wind farm, so for wind directions where it is upstream of the wind farm it likely experiences a wind speed reduction which will be included in the estimated farm wake loss.*

**Answer to R3.16** Indeed, when the winds are from the prevailing direction, there is global blockage effect. However, we expect it to be present in both the mast measurements and mesoscale modeled data. We updated the text to use "wind farm effect" instead of "wind farm wake effect"

to indicate that we also consider a global blockage effect in our analysis and not just a wake effect. The updated text, including comments from reviewer 1 now reads The wind farm effect for the entire October 2013 at the mast can be derived by subtracting the WS-bias of NWF from the WS-bias of EWP or FIT, respectively. Doing so, indicates a wind farm effect at 57 m height, i.e. 12 m below hub height, of 0.2 ms$^{-1}$ for EWP and 0.36 ms$^{-1}$ for FIT. These rather small wind farm effects can be explained by the location of the mast at the west of Rødsand II (Fig. 2), and the relative frequent wind direction for October 2013 from south-west (Fig. 8), when the mast is not much influenced by the wake of the farm, but only by global blockage. Performing the same analysis filtered for eastern wind directions (90°±10°) at the RØ mast, but without filtering for wind speed, indicates a considerable larger wind farm effect based on 127 10-minute values: $WS_{\text{bias,NWF}}$ =1.04 ms$^{-1}$, $WS_{\text{bias,EWP}}$ =0.32 ms$^{-1}$ and $WS_{\text{bias,FIT}}$ =-0.14 ms$^{-1}$, which amounts to a wind farm effect of 0.73 ms$^{-1}$ for EWP and 1.2 ms$^{-1}$ for FIT at 57 m height that can be largely attributed to the wake effect of the farms.. For a more fair comparison, we decided to interpolate the WRF results to the location of the mast, instead of using the nearest grid point, which is either just outside or inside the farm. The updated figure is very similar to the previous figure:

[Figure]

**Comment R3.17** *Line 246 – I think Figure 7 is a better reference than Figure 8.*
**Answer to R3.17** Indeed, we have changed the reference.

**Comment R3.18** *Line 249 – "based on 127 10-minute values". Table 6 says there are 19 periods with wind direction between 80 and 100 degrees and wind speeds between 9 and 12 m/s. Are the 127 m/s 10-minute values including wind speeds outside the 9-12 m/s range?*
**Answer to R3.18** We agree that referencing table 6 here is quite misleading. We only meant to refer to Table 6 for the wind direction filter of the f2 method. In contrast to the flow case analysis, we chose to include wind speeds outside the 9–12 m/s range as you correctly assumed. We removed the reference, so the text now reads Performing the same analysis filtered for eastern wind directions (90°±10°) at the RØ mast, but without filtering for wind speed, indicates ...

**Comment R3.19** *Figures 10-12 – These are hard to read, the features are quite small. Consider making them larger for example by re-arranging the subplots into a 3x2 array instead of the 2x3 used, and by making the full page width.*

**Answer to R3.19** Thank you for the suggestion. For Figure 10, we followed your suggestion of rearranging the subplots. Considering the comment by reviewer 1, regarding Figure 11-12, we decided to introduce sub-figures instead. For Figure 11-12, we also increased the vertical extend and hope that the readability has improved now. We also added error-bars around the SCADA data to show the variability.

[Figure]

[Figure]

**Comment R3.20** *Figure 10 – Consider making the SCADA (f1) and SCADA (f2) dots different colours to better tell them apart.*

**Answer to R3.20** We followed your suggestion and used green for f2 (see answer to previous question).

**Comment R3.21** *Line 267 – I find it difficult to see from Figure 10 that SCADA (f2) agrees better with RANS-ABL than SCADA (f1). Consider making a separate plot showing only the RANS predictions at the turbine positions together with the SCADA data to show this or quantify the deviation.*

**Answer to R3.21** Thanks for this suggestion. We quantified the deviation in the text: "The average RMSE for f1 is 1.1 m/s, while it is 0.4 m/s for f2."

**Comment R3.22** *Line 269 – "indicate" is there twice.*
**Answer to R3.22** Thanks for spotting. We removed one "indicate".

**Comment R3.23** *Line 270 – Lolland is not identified on the map.*
**Answer to R3.23** Indeed. Since it does not play a big role in this analysis. We decided to leave out this geographical term and write instead "the land area to the north of the farms".

**Comment R3.24** *Figure 14 – what data are used for the histograms? Do they include all wind directions and wind speeds?*
**Answer to R3.24** The section 3.3 refers to the f1 filter method as stated at the beginning of the section starting in line 306 of the old manuscript. To make that clear again in the context of the figure description, we added for f1 in line 321 of the old manuscript.

**Comment R3.25** *Figure 15 – Include a description of the vertical dashed lines in the figure caption. Also, the bottom x-axis gives distances relative to some unknown point. Either specify in the caption what zero corresponds to or make the axis relative to the same origin as the top x-axis.*
**Answer to R3.25** We added a legend for the dashed lines. The x-axis is in utm coordinates. However, we agree with the reviewer that it would make more sense to use the same origin for both x-axes. This is done in the updated figure

[Figure]

**Comment R3.26** *Line 363 – Are you trying to say that even far west of Rødsand II the NYRØ and RØ scenarios differ in wind speed, meaning the effect of Nysted on the flow can be still be seen at the western edge of the plot? Consider rephrasing the sentence.*
**Answer to R3.26** Yes, that was exactly what we tried to say. Sorry to not be clear enough. We used your suggestion and modified the text as "However, even far west of Rødsand II the NYRØ and RØ scenarios differ in wind speed, meaning the effect of Nysted on the flow can be still be seen at the western edge of the figure, i.e 27.5 km downstream of Nysted."

**Comment R3.27** *Line 375 – Are you trying to say that wind resources are affected by wind farms more than 25 km away according to the RANS and WRF simulations? Consider rephrasing the sentence.*
**Answer to R3.27** Yes, again that was exactly what we tried to say. We modified the text according to your suggestion "This shows that wind resources are affected by wind farms more than

**Comment R3.28** *Line 378 – Instead of point measurements I suggest snapshots.*
**Answer to R3.28** We modified the text accordingly.

**Comment R3.29** *Line 436 – constraint → constrained.*
**Answer to R3.29** We changed the manuscript accordingly.

**Comment R3.30** *Line 453 – consider adding a reference to van der Laan, P., Peña, A., Volker, P., Hansen, K. S., Sørensen, N. N., Ott, S., & Hasager, C. B. (2017). Challenges in simulating coastal effects on an offshore wind farm: Paper. Journal of Physics: Conference Series, 854, [012046].*
**Answer to R3.30** We added the reference as follows following for instance the approach in van der Laan et al. (2017).

**References**

Fitch, A. C., Olson, J. B., Lundquist, J. K., Dudhia, J., Gupta, A. K., Michalakes, J., and Barstad, I.: Local and Mesoscale Impacts of Wind Farms as Parameterized in a Mesoscale NWP Model, Monthly Weather Review, 140, 3017–3038, https://doi.org/10.1175/MWR-D-11-00352.1, URL `http://www.vattenfall.co.uk/en/thanet-`, 2012.

Gaumond, M., Réthoré, P.-E., Ott, S., Peña, A., Bechmann, A., and Hansen, K. S.: Evaluation of the wind direction uncertainty and its impact on wake modeling at the Horns Rev offshore wind farm, Wind Energy, 17, 1169, 2014.

Hansen, K. S., Réthoré, P.-E., Palma, J., Hevia, B. G., Prospathopoulos, J., Peña, A., Ott, S., Schepers, G., Palomares, A., van der Laan, M. P., and Volker, P.: Simulation of wake effects between two wind farms, Journal of Physics: Conference Series, 625, 012 008, https://doi.org/10.1088/1742-6596/625/1/012008, URL `https://iopscience.iop.org/article/10.1088/1742-6596/625/1/012008`, 2015.

Nygaard, N. G. and Hansen, S. D.: Wake effects between two neighbouring wind farms, Journal of Physics: Conference Series, 753, 032 020, https://doi.org/10.1088/1742-6596/753/3/032020, URL `https://iopscience.iop.org/article/10.1088/1742-6596/753/3/032020`, 2016.

Panofsky, H. A. and Dutton, J. A.: Atmospheric Turbulence, Wiley-interscience, 1984.

Poulsen, L.: 1.7_Poulsen: Validation of wind farm parametrisation in WRF using wind farm data, Tech. rep., DTU, https://doi.org/10.5281/ZENODO.3637944, URL `https://doi.org/10.5281/zenodo.3637944#.YbtIv61bN8k.mendeley`, 2019.

van der Laan, M. P., Sørensen, N. N., Réthoré, P.-E., Mann, J., Kelly, M. C., Troldborg, N., Hansen, K. S., and Murcia, J. P.: The $k$-$\varepsilon$-$f_P$ model applied to wind farms, Wind Energy, 18, 2065–2084, https://doi.org/10.1002/we.1804, URL `https://onlinelibrary.wiley.com/doi/10.1002/we.1804`, 2015.

van der Laan, M. P., Peña, A., Volker, P., Hansen, K. S., Sørensen, N. N., Ott, S., and Hasager, C. B.: Challenges in simulating coastal effects on an offshore wind farm, Journal of Physics: Conference Series, 854, 012 046, https://doi.org/10.1088/1742-6596/854/1/012046, URL `https://iopscience.iop.org/article/10.1088/1742-6596/854/1/012046`, 2017.

---

## Author Response (AR2)

**Response to the comments about the submitted paper**

**Comparing and validating intra-farm and farm-to-farm wakes across different mesoscale and high-resolution wake models**

We would like to thank Referee #3, Nicolai Gayle Nygaard, for the careful reading of our revised manuscript and for the useful comments and suggestions. Our detailed answers follow.

Please note that reviewers' comments are in italics while our answers are not. Additions to the original manuscript are indicated in blue.

**Answers to Reviewer 3**

**Comment R3.1** - *p. 5 where does the pitch curve come from (eg turbine manufacturer or measured using a separate wind speed reference). Consider including a plot of it in the paper*
**Answer to R3.1** Thank you for that question. We checked the details of the derivation of the equivalent wind turbine wind speed again and found that we only relied on the power curve below rated wind speed. Thus, we did not use the pitch curve at all. Sorry about the confusion. We modified the text, which now reads "The SCADA data include electric power, rotor speed, yaw position and nacelle wind speed.The SCADA data has been quality-controlled (Hansen et al., 2015). From the SCADA data, the equivalent wind turbine wind speed was derived from the 10-minute values of the power combined with the power curve below rated power.".

**Comment R3.2** *Line 126 - I don't think the abbreviation RØ has ben defined before it is used*
**Answer to R3.2** Thanks for spotting this. We have replaced "RØ" by "Rødsand II". In addition, we defined RØ and NY at line 70 of the new version: "The Fehmarn Belt with the wind farms Nysted (abbreviated NY in the following) and Rødsand II (abbreviated RØ in the following) has been selected as the study area (Fig. 2)".

**Comment R3.3** *Fig. 6+8 - The abbreviation RS2-mast conflicts with the text where the mast is called RØ mast*
**Answer to R3.3** We changed the title of figure 6. The y-label of figure 8 has been changed already (see Answer to R3.12 provided in response to the first review), but we accidentally included the old figure in the revised manuscript.

**Comment R3.4** *Line 224 - "averaged over" is repeated*
**Answer to R3.4** We removed one "averaged over". Additionally we removed one "filter" in line 332 and one "for" in line 400 of the submitted revised manuscript with tracked changes.

**Comment R3.5** *Fig. 9 caption - Change "interpolated WRF results to" to "WRF results interpolated to"*
**Answer to R3.5** Changed.

**Comment R3.6** *Fig. 9 caption - change time steps to time stamps*
**Answer to R3.6** Changed.

**Comment R3.7** *Line 296 - can the larger bias not also be due to global blockage which is included in EWP and FIT, but not in NWP?*
**Answer to R3.7** Global blockage should be embedded in the mast measurements as well as in the EWP and FIT simulations, but not in the NWF simulations. Thus, one would expect EWP and FIT to be more accurate than NWF. However, we don't find this from our simulations ($WS_{\mathrm{bias,NWF}}$ =-0.02 ms$^{-1}$, $WS_{\mathrm{bias,EWP}}$ =-0.17 ms$^{-1}$ and $WS_{\mathrm{bias,FIT}}$ =-0.28 ms$^{-1}$). The reason for that is that a simple linear interpolation to the mast location using grid points within the farm and outside the farm cannot present the wind conditions at the mast accurately enough. This has been explained in the text as "The larger biases for EWP and FIT compared to NWF for the non-wake-affected sector can be partly attributed to the interpolation of wake-affected and non-wake-affected points to the mast location". The error due to the interpolation exceeds the effect of missing global blockage in NWF in this case.

**Comment R3.8** *Line 323 - change taken to handled*
**Answer to R3.8** Changed.

**Comment R3.9** *Line 325 - add filtering before methods*
**Answer to R3.9** Changed.

**Comment R3.10** *Line 326 - delete rotor in rotor equivalent wind speed to be consistent with the rest of the text and the literature in general*
**Answer to R3.10** Changed.

**Comment R3.11** *Line 384 - ZON is the most complex engineering wake model in this study, not in general :-)*
**Answer to R3.11** Agreed. We added "used in this study" to the text.

**Comment R3.12** *Line 399 - add an s on deficit*
**Answer to R3.12** Changed.

**Comment R3.13** *Line 404 - should it be they?*
**Answer to R3.13** Indeed, we changed it accordingly.

**Comment R3.14** *Line 439 - change the to a distance of before 5 rotor diameters*
**Answer to R3.14** Changed.

**Comment R3.15** *Line 471 - add the before baseline*
**Answer to R3.15** Changed.

**Comment R3.16** *Line 481 - do the authors believe that engineering models should be calibrated differently for intra-farm and farm-farm interactions? In that case, how would one use such models in situations where neighbouring wind farms are present?*
**Answer to R3.16** Thank you for that comment. We did not intend to argue for a different calibration for intra-farm and farm-farm interaction for engineering models. Instead we propose to also include farm-to-farm data sets in addition to intra-farm data sets for calibration. To make this clear, we modified the text as follows: "Future work could consider farm-to-farm calibration data-sets in addition to intra-farm calibration data-sets to improve their capability."

**Comment R3.17** *Line 508 - a wind speed before variability*
**Answer to R3.17** Changed.